# Chiral multiferroicity in two-dimensional hybrid organic-inorganic perovskites

Haining Zheng [1,2,10], Arup Ghosh[3,10], M. J. Swamynadhan[4,10], Qihan Zhang [5], Walter P. D. Wong[2], Zhenyue Wu[2], Rongrong Zhang[2], Jingsheng Chen [5], Fanica Cimpoesu[6], Saurabh Ghosh[4], Branton J. Campbell [7], Kai Wang [8] ✉, Alessandro Stroppa [9] ✉, Ramanathan Mahendiran [3] ✉ & Kian Ping Loh [1,2] ✉

Chiral multiferroics offer remarkable capabilities for controlling quantum devices at multiple levels. However, these materials are rare due to the competing requirements of long-range orders and strict symmetry constraints. In this study, we present experimental evidence that the coexistence of ferroelectric, magnetic orders, and crystallographic chirality is achievable in hybrid organic-inorganic perovskites $[(R/S)\text{-}\beta\text{-methylphenethylamine}]_2CuCl_4$. By employing Landau symmetry mode analysis, we investigate the interplay between chirality and ferroic orders and propose a novel mechanism for chirality transfer in hybrid systems. This mechanism involves the coupling of non-chiral distortions, characterized by defining a pseudo-scalar quantity, $\xi = \mathbf{p} \cdot \mathbf{r}$ ($\mathbf{p}$ represents the ferroelectric displacement vector and $\mathbf{r}$ denotes the ferro-rotational vector), which distinguishes between $(R)$- and $(S)$-chirality based on its sign. Moreover, the reversal of this descriptor's sign can be associated with coordinated transitions in ferroelectric distortions, Jahn-Teller antiferro-distortions, and Dzyaloshinskii-Moriya vectors, indicating the mediating role of crystallographic chirality in magnetoelectric correlations.

Chirality, the property of non-superposability between an object and its mirror image, is of paramount importance in research fields ranging from life chemistry to materials science. As a form of structure-inversion spatial asymmetry, chirality enables the Dresselhaus-Rashba-type spin-orbit coupling (SOC) and the topological confined spin arrangements[1,2], which engenders rich magnetic phenomena, such as chiral skyrmions[3,4], persistent spin textures[5,6], and Dzyaloshinskii-Moriya (D-M) interaction[7,8]. Furthermore, chiral crystals, with naturally reduced symmetry, tend to adopt chiral polar point groups $C_1$, $C_2$, $C_3$, $C_4$, or $C_6$[9], which offers a pathway to induce polar electric long-range orders. Another peculiar feature is the chirality-induced spin selectivity (CISS), which allows the manipulation of spin polarization without external magnetic fields, thus providing potential for spin filters and non-volatile magnetic memories[10,11]. This highlights the fact that chirality may serve as another degree of freedom to couple electric and magnetic orders. However, the design of single-phase chiral multiferroics is challenging because the ferroelectric and magnetic orders are often mutually exclusive, which require empty and partially filled

[1]Joint School of National University of Singapore and Tianjin University, International Campus of Tianjin University, Binhai New City, Fuzhou 350207, China. [2]Department of Chemistry, National University of Singapore, 3 Science Drive 3, Singapore 117543, Singapore. [3]Department of Physics, National University of Singapore, 2 Science Drive 3, Singapore 117551, Singapore. [4]Department of Physics and Nanotechnology, SRM Institute of Science and Technology, Kattankulathur 603203 Tamil Nadu, India. [5]Department of Materials Science and Engineering, National University of Singapore, Singapore 117575, Singapore. [6]Institute of Physical Chemistry, Splaiul Independentei 202, Bucharest 060021, Romania. [7]Department of Physics & Astronomy, Brigham Young University, Provo, UT 84602, USA. [8]Key Laboratory of Luminescence and Optical Information, Ministry of Education, School of Physical Science and Engineering and Institute of Optoelectronics Technology, Beijing Jiaotong University, Beijing 100044, China. [9]CNR-SPIN, c/o Dip.to di Scienze Fisiche e Chimiche - University of L'Aquila, Via Vetoio, Coppito (AQ), 67100, Italy. [10]These authors contributed equally: Haining Zheng, Arup Ghosh, M. J. Swamynadhan. ✉e-mail: kaiwang@bjtu.edu.cn; alessandro.stroppa@spin.cnr.it; phyrm@nus.edu.sg; chmlohkp@nus.edu.sg

$d$-orbital occupancies, respectively[12]. The rarity of chiral multiferroics is also restricted by the low structural symmetry of chiral polar point groups.

One ideal material paradigm in this regard is layered $Cu^{2+}$-based hybrid organic-inorganic perovskites (HOIPs)[13–15]. This class of compounds exhibits corner-sharing $CuCl_6$ octahedral frames connected with A$^+$-site organic cations through supramolecular hydrogen bonding. Compared to prototypical inorganic oxide perovskites, the ferroelectricity in HOIPs stems from the disorder-to-order phase transition of organic ligands and the induced structural distortions into the inorganic octahedra[16], while the cooperative Jahn-Teller (J-T)-tilted ordering provides the ferromagnetic superexchange pathway along $Cu^{2+}$-$Cl^-$-$Cu^{2+}$ bridges[15], thereby affording the multiferroic nature. Besides, the incorporation of chiral organic ligands enables the chirality transfer across the organic-inorganic framework, which is synthetically coupled with the helical distortion of $CuCl_6$ anionic cages[17]. Despite a few examples of HOIPs-based multiferroics[18–20], the intricate interplay with structural chirality transfer and multiple ferroic orders in HOIPs has yet to be revealed. The critical concerns are whether the crystallographic chirality is compatible with the multiferroic behaviors in HOIPs and how the chirality order parameters are cross-coupled with their intrinsic ferroic orders.

Herein, we demonstrate the coexistence of chirality, electric and magnetic orders within HOIP-based chiral multiferroics (R/S)-(MPA)$_2$CuCl$_4$ (MPA = $\beta$-methylphenethylamine), which display intralayer ferroelectricity and A-type antiferromagnetic order. These HOIPs also exhibit chirality-dependent magnetic circular dichroism (MCD) characters arising from the field-induced Zeeman effect. Through the Landau-type symmetry-mode analysis, we reveal a peculiar chirality transfer mechanism, wherein two non-chiral structural distortions hybridize to break all the improper symmetries $S_n$ in the structure, thereby rendering structural chirality. This chirality transfer mechanism can be parametrized by a pseudo-scalar order parameter, $\xi = \mathbf{p} \cdot \mathbf{r}$, ($\mathbf{p}$ is the ferroelectric moment and $\mathbf{r}$ is the ferro-rotational moment,

which are non-chiral order parameters), wherein the sign of $\xi$ manifests as +1 and −1 for the (R)- and (S)-chiral HOIPs, or vice versa. In addition, the chirality change is allowed by symmetry to couple with J-T pseudo-rotations of the associated orbital ordering as well as the D-M vectors, thereby allowing the possibility of a synergetic correlation between the intralayer ferroelectric and ferromagnetic moments[21,22]. This study highlights the correlation of crystallographic chirality with the ferroic behaviors in HOIPs, paving the ground for studying other spin-related properties such as chiral spin textures and chiral magneto-optical effects.

## Results

### Crystal structures of chiral copper perovskites

To examine the effect of organic-to-inorganic chirality transfer on lattice distortions and its correlation with magnetic and electric orders, we synthesized the chiral copper-based perovskite single crystals (R/S)-(MPA)$_2$CuCl$_4$ using a facile solvent evaporation crystallization method in the atmospheric environment. The powder-formed racemic counterpart (MPA)$_2$CuCl$_4$ was also prepared using the same synthesis method as the control (the detailed synthesis procedures can be found in the "Methods" section). Fig. 1a depicts the top and side views of crystallographic structures of (R/S)-(MPA)$_2$CuCl$_4$. Both (R)- and (S)-form perovskite enantiomers possess 2D Ruddlesden–Popper-type layered configurations, with corner-sharing $CuCl_6$ octahedral inorganic frames intercalated by bilayers of organic spacers [(R/S)-MPA]$^+$ via N−H⋯Cl hydrogen bonding interaction (Fig. 1b), forming a natural "quantum-well" configuration[23]. Alternating framework layers are offset from one another along in-plane crystal direction by half a nearest-neighbor $Cu^{2+}$-$Cu^{2+}$ distance. The phase purity and crystallization of perovskite single crystals have been confirmed by powder X-ray diffraction (XRD) measurements, which display remarkable long-term chemical stability that does not show any evident decomposition after one year in an ambient environment (see Fig. 1c and Supplementary Fig. 1).

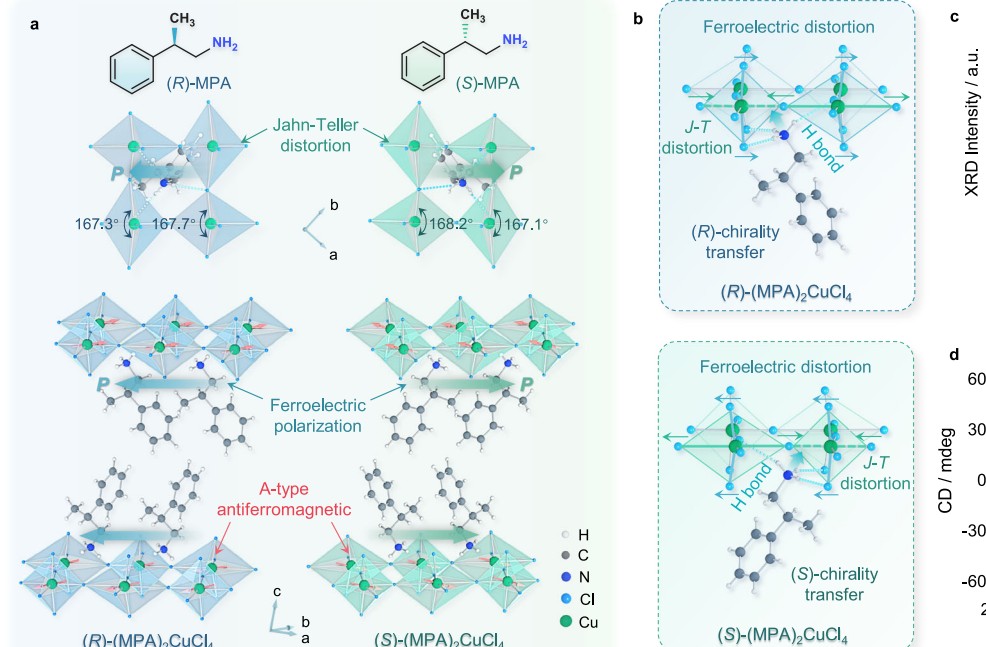

**Fig. 1 | Crystallographic structures and magnetic circular dichroism of (R/S)-(MPA)$_2$CuCl$_4$. a** The crystallographic structures of (R)- and (S)-(MPA)$_2$CuCl$_4$ (MPA = $\beta$-methylphenethylamine) with opposite polarization directions and A-type antiferromagnetic configurations, as well as the top views that show the Jahn-Teller (J-T) distorted patterns. **b** Schematic diagrams of organic-to-inorganic chiral

transfer and lattice distortion in (R/S)-(MPA)$_2$CuCl$_4$. **c** Powder X-ray diffraction (XRD) and corresponding simulated XRD patterns for (R/S)-(MPA)$_2$CuCl$_4$. **d** Magnetic circular dichroism (MCD) for (R/S)-(MPA)$_2$CuCl$_4$ under the external magnetic fields of 1.6 T, 0 T, and -1.6 T along the $H_{ext} \parallel c$-axis direction.

The incorporation of chiral organic cations [(R/S)-MPA]$^+$ allows the chirality transfer to the inorganic metal-halide sublattice via hydrogen bonding interaction and permits (R/S)-(MPA)$_2$CuCl$_4$ to crystallize in space group of P1 (polar point group C1) (see Supplementary Table 1 and Supplementary Table 2). Such chiral crystallographic structures render (R/S)-(MPA)$_2$CuCl$_4$ enantiomers display isostructural mirrored configurations, wherein all of the polar [(R/S)-MPA]$^+$ cations with permanent dipoles are uniformly aligned along [1 1 0] or [-1 -1 0] direction, leading to the shift of positive charge centers (Fig. 1a, bottom). The inorganic CuCl$_6$ anionic cages also display chirality-induced octahedral deformation with mirrored tilts of apical Cl$^-$ ions and J-T antiferro-distortion. As depicted in Fig. 1b, the J-T-active Cu$^{2+}$ cations induce the CuCl$_6$ octahedral cages to possess alternately elongated and compressed Cu$^{2+}$-Cl$^-$ bonds with an antiferro-distortive arrangement along the ab plane, which is mirrored for (R)- and (S)-form enantiomers along [1 1 0] direction. Such structural distortion of CuCl$_6$ octahedra is also accompanied by the small tilts of Cl$^-$-Cu$^{2+}$-Cl$^-$ bond angles away from the standard value of 180°, resulting in the off-centering displacement of negative charges. Therefore, the cooperative distortions of organic cations and inorganic cages separate the positive and negative charge centers of (R/S)-(MPA)$_2$CuCl$_4$, giving rise to an opposite spontaneous polarization along [1 1 0] or [-1 -1 0] direction.

## Magnetic anisotropy

A recent report by Sun et al. revealed that the powder crystals of (R/S)-(MPA)$_2$CuCl$_4$ exhibit soft ferromagnetic properties, with a Curie temperature $T_c$ of approximately 6 K[15]. Herein, we focus on the high-quality single-crystal samples and examine the magnetic anisotropy by characterizing the magnetic hysteresis loops along the a, b, and c-axis directions, respectively. As presented in Fig. 2a, e, the field-dependent magnetization (M-H) curves present a characteristic soft ferromagnetic behavior in the CuCl$_6$ plane ($H_{ext} \perp c$-axis), and reach the saturated magnetization $M_s$ of 1 $\mu_B$/f.u. at 4000 Oe, consistent with the Cu$^{2+}$ spin magnetic moment (S = 1/2). The M-H curves of racemic (MPA)$_2$CuCl$_4$ shown in Supplementary Fig. 2 also display ferromagnetic features. In addition, as plotted in Supplementary Fig. 3, the magnetic moments for both (R)- and (S)-form enantiomers are comparable when measured along a-axis and b-axis, i.e., the magnetic

anisotropy is negligible along the in-plane direction. By contrast, a clear antiferromagnetic transition occurs at T = 4 K when the magnetic field is applied perpendicular to the CuCl$_6$ plane ($H_{ext} \parallel c$ axis); it can be easily converted to ferromagnetic ordering under higher external magnetic fields (>1 kOe) (Fig. 2b, f). In contrast to the ferromagnetic behavior observed in Sun's work, our findings indicate that (R/S)-(MPA)$_2$CuCl$_4$ display intra-layer ferromagnetic and inter-layer antiferromagnetic properties, i.e., a typical A-type antiferromagnetic configuration. Such a pattern is associated with the geometrical anisotropy of stratified Ruddlesden–Popper-type configuration, where the magnetic CuCl$_6$ frame is separated by non-magnetic (R/S)-(MPA)$^+$ cations along the c-axis[24]. The intra-layer ferromagnetic behavior is also in agreement with the Goodenough–Kanamori rules. This is a consequence of the orbital orthogonality of the magnetic Cu$^{2+}$ 3d orbitals[15,25], induced by the alternate orientation (mutually perpendicular) of the J-T elongation axes within the 2D framework[22,24].

## Magnetic circular dichroism

The coexistence of optical activity (NOA) and ferromagnetism in (R/S)-(MPA)$_2$CuCl$_4$ allows us to explore magneto-chiroptical effects in these crystals. The dielectric functions $\varepsilon_\pm$ (the sign ± indicates left and right chirality) can be decomposed into several components: the field-independent dielectric function, which does not contribute to the chiroptical effects, the natural circular dichroism (NCD) resulting from the space inversion asymmetry, the MCD arising from the breaking of time-reversal symmetry, and the magneto-chiral dichroism (MChD) originating from the correlation of MCD and NCD[15,26]. A recent study on MChD demonstrated that (R/S)-(MPA)$_2$CuCl$_4$ exhibits pronounced and mirror-imaged MChD signals at 2 K, highlighting a strong correlation between crystallographic chirality and intrinsic magnetism[15]. We characterized the dissymmetry factor of MChD ($g_{MChD}$) for polycrystalline thin films (Supplementary Fig. 4). The corresponding $g_{MChD}$ values are summarized in Supplementary Table 3. The MChD effect stems from the interference of electric dipole and magnetic dipole transitions, as well as the Cotton-Mouton effect due to the second-order perturbation of the magnetic field between excited states[26]. The values are higher than those reported for Eu((±)tfc)$_3$ complex (5 × 10$^{-3}$ T$^{-1}$)[27] and chiral Ni nanomagnets (7.3 × 10$^{-4}$ T$^{-1}$)[28].

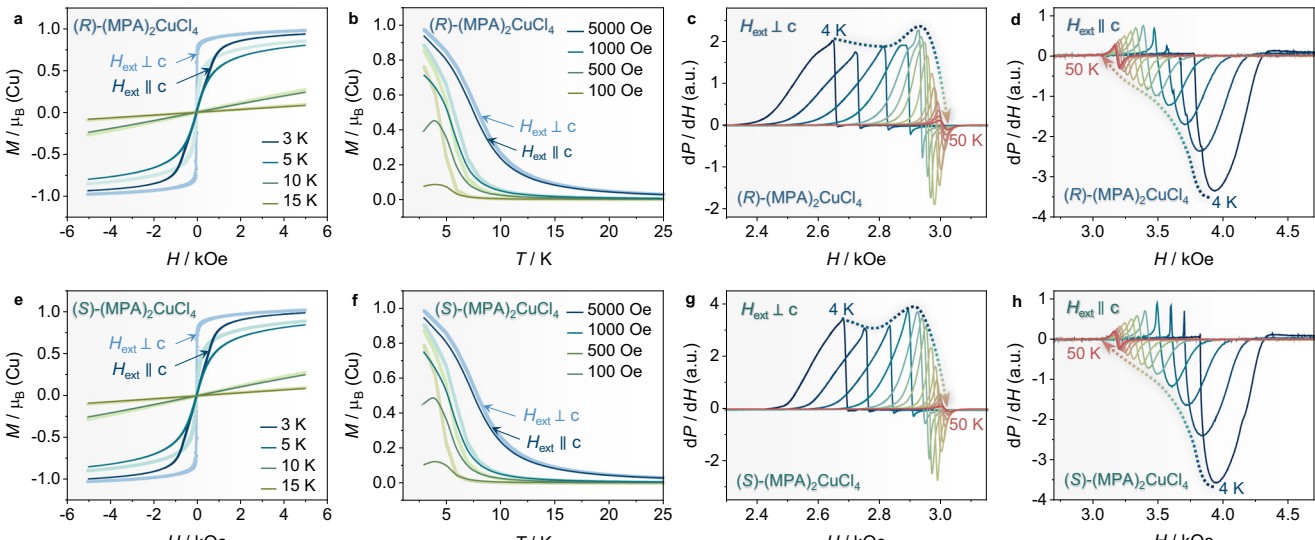

**Fig. 2 | Magnetic anisotropy and ferromagnetic resonance (FMR) in (R/S)-(MPA)$_2$CuCl$_4$. a, e** Field-dependent magnetization (M-H) curves of (R/S)-(MPA)$_2$CuCl$_4$ when the external fields are applied parallel ($H_{ext} \perp c$-axis) or perpendicular ($H_{ext} \parallel c$-axis) to CuCl$_6$ plane. **b, f** Temperature-dependent magnetization (M-T) curves of (R/S)-(MPA)$_2$CuCl$_4$ when the external fields are applied parallel

($H_{ext} \perp c$-axis) or perpendicular ($H_{ext} \parallel c$-axis) to CuCl$_6$ plane. **c, g** Field (H) – sweep FMR spectra at the fixed frequency of 9 GHz for (R/S)-(MPA)$_2$CuCl$_4$ when the external fields are applied parallel ($H_{ext} \perp c$-axis) to the CuCl$_6$ plane. **d, h** Field (H) – sweep FMR spectra at the fixed frequency of 9 GHz for (R/S)-(MPA)$_2$CuCl$_4$ when the external fields are applied perpendicular ($H_{ext} \parallel c$-axis) to CuCl$_6$ plane.

Furthermore, MCD studies were performed by placing the (R/S)-(MPA)$_2$CuCl$_4$ polycrystalline thin films between a pair of permanent magnets with maximum fields of ± 1.6 T. The field direction could be adjusted in parallel or antiparallel directions with respect to an incoming light wave vector. As displayed in Fig. 1d, the (R)- and (S)-(MPA)$_2$CuCl$_4$ exhibit mirrored and derivative-like lineshapes of CD peaks at 220 nm and 321 nm, which is in accordance with the characteristic of 'Cotton effect'[29]. The anisotropy factors (g$_{CD}$) (defined as |g$_{CD+}$ − g$_{CD-}$|/2)) for (R)- and (S)-(MPA)$_2$CuCl$_4$ are estimated to be approximately 0.0038 and 0.0029 at 223 nm respectively, which are much larger than chiral bismuth hybrids (R/S)-(C$_8$H$_{12}$N)$_4$Bi$_2$Br$_{10}$ and comparable with 2D lead iodide perovskites[30,31]. By comparing with the CD spectra for H = 0, the (R)- and (S)-(MPA)$_2$CuCl$_4$ respond oppositely to the magnetic field directions. The increase of the field up to 1.6 T enhances the CD strength at 321 nm near the absorption band-edge for the (R)-(MPA)$_2$CuCl$_4$, whereas it diminishes the CD strength for its (S)-form counterpart. Such MCD responses are analogous to those observed in prototypical 2D chiral HOIPs (R/S)-(methylbenzylamine)$_2$PbI$_4$ polycrystalline films, which exhibit comparable MCD strengths at 1.6 T[32]. The effect originates from the fine-tuning of the chiral-dependent exciton energies by the field-induced Zeeman splitting[33]. In (R)- and (S)-(MPA)$_2$CuCl$_4$, the excitonic transitions display dependence on chirality and can be accessed by left or right-circularly polarized photoexcitation, respectively[32].

## Ferromagnetic resonance

Broadband ferromagnetic resonance (FMR) spectra were also recorded to investigate the magnetic anisotropy of (R/S)-(MPA)$_2$CuCl$_4$. Figure 2c, d, g, h display the field-dependent FMR spectra from 4 K to 50 K, plotted as the field derivative of power absorption (dP/dH) as the function of magnetic field (H) at a fixed frequency of 9 GHz. When the magnetic field is applied along the ab plane (H$_{ext}$ ⊥ c-axis) (Fig. 2c, g), both (R)- and (S)-(MPA)$_2$CuCl$_4$ exhibit positive dP/dH peaks with negligible negative components in the ferromagnetic regime (<8 K). As the temperature rises to the paramagnetic regime (>8 K), FMR spectra gradually change to a Lorentzian-type antisymmetric lineshape, accompanied by a decline of dP/dH peak magnitude and a shift of resonance field (H$_r$) towards high fields. Notably, the FMR spectra present opposite signals of dP/dH peaks and an inverse shift of H$_r$ with temperature when the magnetic field is parallel to the c-axis (H$_{ext}$ ∥ c-axis), as shown in Fig. 2d, h. This inverse variation of FMR spectra along intra-layer and inter-layer directions resembles the behavior observed in 2D van der Waals magnets (e.g. CrCl$_3$ and Fe$_5$GeTe$_2$)[34–36]. This phenomenon can be attributed to the strong magnetocrystalline anisotropy caused by D-M interaction-induced spin canting in layered lattice frame, implying that the magnetic easy axis lies in the intra-layer ab plane[36,37]. Furthermore, as plotted in Supplementary Fig. 5 and Supplementary Fig. 6, the resonance field H$_r$ (defined as the zero-crossing point of FMR spectra) presents a continuous shift with the temperature, which can be described by the Kittel equation

$$f_r = \frac{\gamma}{2\pi}\sqrt{H_r(H_r + 4\pi M_{eff})} \tag{1}$$

wherein $M_{eff}$ is the effective magnetization and $\gamma$ is the gyromagnetic ratio ($\gamma = \frac{g\mu_B}{\hbar}$, g is the Landé g-factor, $\mu_B$ is the Bohr magneton, and $\hbar$ is the reduced Planck's constant). At temperatures above the T$_c$, the fitting of experimental data at 50 K yields Landé g-factors of 2.171 (2.047) and 2.138 (2.052) for (R)- and (S)-(MPA)$_2$CuCl$_4$ along the intra-layer (inter-layer) direction (50 K), which is slightly larger than the free-electron g-factor of 2.0023. These anisotropic Landé g-factors arise from the orbital moment induced by magnetocrystalline anisotropy that affects the preferred orientation of Cu$^{2+}$ magnetic moments through SOC effect[35].

In the paramagnetic regime (>20 K), FMR spectra exhibit well-defined antisymmetric lineshapes, which can be nicely fitted using the derivative Lorentzian function (Supplementary Figs. 7 and 8); this signifies the homogeneous magnetic states of Cu$^{2+}$ spins. Thus, the spin decoherence lifetime $\tau$ of chiral HOIPs can be estimated through the relation $\tau = \frac{2}{\sqrt{3}\gamma\Delta H_{pp}} = \frac{4\pi}{\sqrt{3}\Delta f_{pp}}$, wherein $\Delta f_{pp}$ and $\Delta H_{pp}$ denote the frequency and field FMR peak-to-peak linewidth, respectively. As plotted in Supplementary Figs. 5 and 6, both (R)- and (S)-(MPA)$_2$CuCl$_4$ display nanosecond-scale $\tau$ of 2.702 ± 0.327 ns (H$_{ext}$ ⊥ c-axis) and 1.590 ± 0.144 ns (H$_{ext}$ ∥ c-axis), which are much longer than typical Pb$^{2+}$- and Sn$^{2+}$-based HOIPs[38–40], and comparable with the magnetic Mn$^{2+}$-based hybrid organic-inorganic analogs[41]. The shorter $\tau$ along the inter-layer direction (H$_{ext}$ ∥ c-axis), i.e., magnetic hard axis, is a consequence of the increased energy barriers for spin alignment compared to the energetically favorable intra-layer direction (H$_{ext}$ ⊥ c-axis)[37].

## Electrically switchable ferroelectric properties

The ferroelectricity of (R/S)-(MPA)$_2$CuCl$_4$ is verified by electric polarization versus voltage (P-V) hysteresis loop measurements using single-crystal samples along the polar axis [1 1 0] or [-1 -1 0] direction (see "Methods" for more details). As displayed in Fig. 3a, both (R)- and (S)-(MPA)$_2$CuCl$_4$ exhibit well-shaped ferroelectric P-V hysteresis loops with spontaneous polarizations P$_s$ of 7.0 and 6.2 μC cm$^{-2}$ respectively, which are almost equal within experimental errors, confirming the electrically switchable polarization properties. By contrast, the racemic (MPA)$_2$CuCl$_4$ only presents linear dielectric characteristics with negligible remanent polarization. The absence of a ferroelectric-to-paraelectric phase transition up to the decomposition temperature of 450 K is observed in the thermogravimetric analysis (TGA) and differential scanning calorimetry (DSC) measurements (Supplementary Fig. 9), indicating that the ferroelectric Curie temperature T$_c$ of (R/S)-(MPA)$_2$CuCl$_4$ is higher than its decomposition temperature, similar to molecular ferroelectrics (R/S)-[Zn(OAc)(L)Yb(NO$_3$)$_2$][42] and [(CH$_3$)$_4$N]HgCl$_3$[43].

To provide the microscopic evidence of ferroelectric domains and electrically switchable properties, we carried out piezo-response force microscopy (PFM) measurements at room temperature[44,45]. The electrically switchable polarization properties of (R/S)-(MPA)$_2$CuCl$_4$ have been confirmed by applying a sweeping voltage of ± 12 V on the surface of perovskite thin films, where the rectangular phase hysteresis loops and butterfly-shaped amplitude curves are observed in Fig. 3b, c. In addition, as depicted in Fig. 3d–i, both (R)- and (S)-(MPA)$_2$CuCl$_4$ exhibit well-defined ferroelectric domains with ~180° phase contrast in phase and amplitude PFM images along the in-plane direction (ab plane), while the PFM response along the out-of-plane direction (c-axis) is negligible (as shown in Supplementary Fig. 10), in accordance well with the crystallographic analysis of uniaxial ferroelectric nature. By contrast, no distinct ferroelectric domain patterns can be observed for racemic (MPA)$_2$CuCl$_4$, consistent with its linear dielectric characteristic. Furthermore, the angular-resolved PFM study of (R/S)-(MPA)$_2$CuCl$_4$ was performed by rotating the sample along the clockwise direction with respect to the PFM cantilever[46]. As demonstrated in Supplementary Figs. 11 and 12, with the increase of rotation angles, the sign of in-plane phase signals changes gradually from negative (the dark contrast in PFM images of 60° and 120°) to positive (the bright contrast in PFM images of 240° and 300°), and the in-plane domain patterns show opposite contrast when the azimuth angular difference is 180° (as highlighted by the white broken circles). Such angular dependence of domain evolution accords well with the cosine law of piezo-response amplitude and rotation angles in uniaxial in-plane ferroelectrics[46], further confirming the intrinsic uniaxial ferroelectric nature of (R/S)-(MPA)$_2$CuCl$_4$.

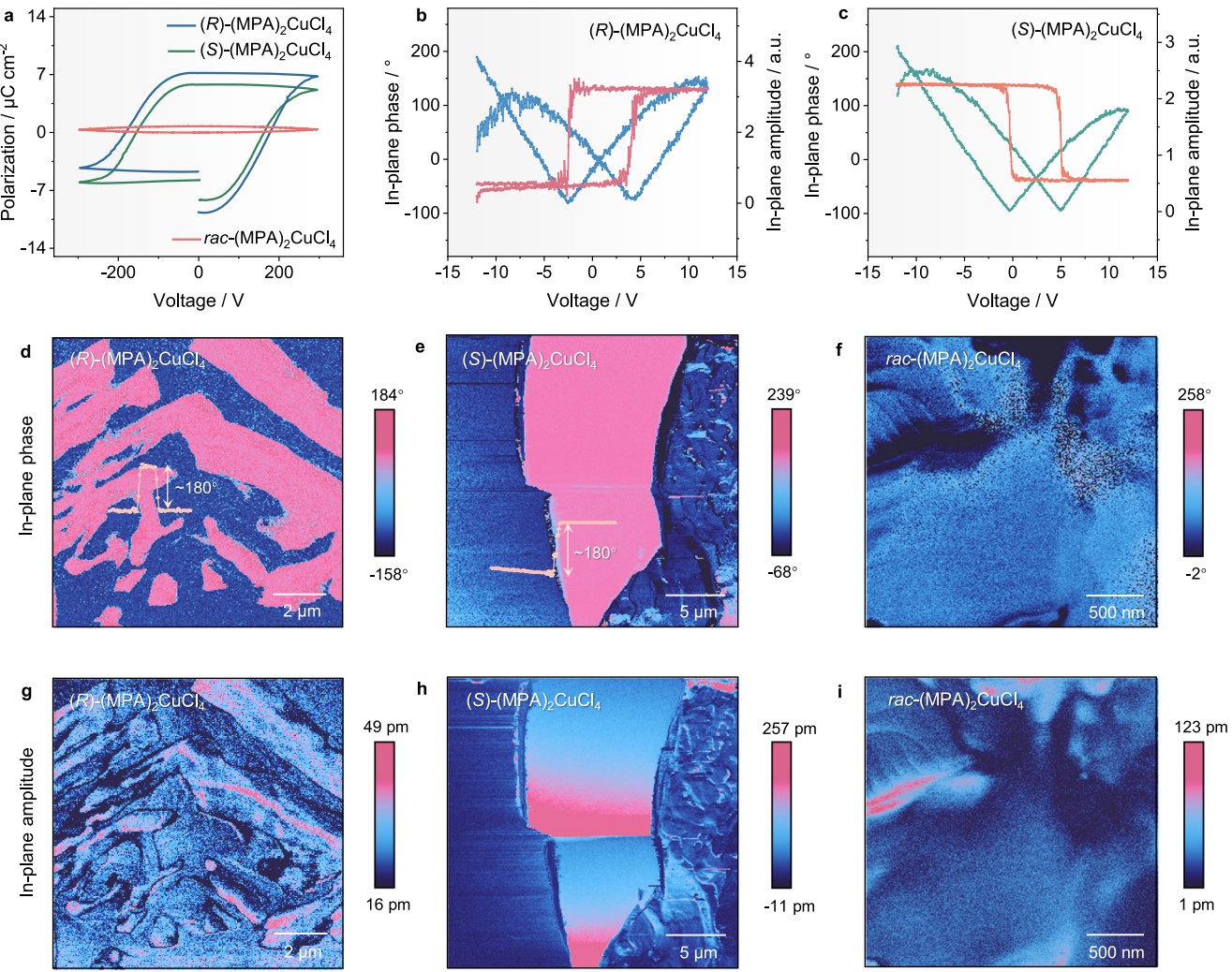

**Fig. 3 | Ferroelectric characterization of $(R/S)$-(MPA)$_2$CuCl$_4$ and their racemic counterpart. a** Ferroelectric polarization – voltage ($P$-$V$) hysteresis loops of $(R/S)$-(MPA)$_2$CuCl$_4$ and their racemic structure as reference. **b** The phase hysteresis loop and amplitude butterfly curves of $(R)$- (MPA)$_2$CuCl$_4$ and **c** $(S)$-(MPA)$_2$CuCl$_4$ along the in-plane direction. **d** In-plane piezoelectric force microscopy (PFM) phase images of $(R)$-(MPA)$_2$CuCl$_4$, **e** $(S)$-(MPA)$_2$CuCl$_4$ and **f** racemic (MPA)$_2$CuCl$_4$. **g** In-plane PFM amplitude images of $(R)$-(MPA)$_2$CuCl$_4$, **h** $(S)$-(MPA)$_2$CuCl$_4$, and **i** racemic (MPA)$_2$CuCl$_4$.

## Theoretical calculations of ferroelectric polarization

To gain insights into the origin of ferroelectric polarization, we have constructed a chirality-preserving transition path by introducing a non-centrosymmetric non-polar reference structure. This structure was constructed by rotating half of the organic molecules by 180° to compensate for their dipole moments and enforcing the spatial inversion symmetry on the CuCl$_4$ framework (see Supplementary Fig. 13, Supplementary Note 1 and Supplementary Note 2 for more details on the computational methods). The symbol $\lambda \in [0,1]$ was introduced to parametrize the chirality-preserving transition path from the non-centrosymmetric non-polar reference structure ($P_0$, $\lambda = 0$) to the polar ground state structure ($P_+$ or $P_-$, $\lambda = \pm1$)[47], as illustrated in Fig. 4a the points along the path are connected by first-kind symmetry operations, i.e., handness preserving operations; therefore, the designed path is homochiral. The intermediate structures were generated as computational configurations to calculate the continuous and smooth evolution of the polarization. The electric polarization is estimated to be ~ 7.8 μC cm$^{-2}$, with a main component along the diagonal direction in the $ab$ plane ($P_{ab}$), consistent with the experimental results. The ferroelectric polarization originates from the cooperative coupling between electric dipole moments of organic cations [$(R/S)$-MPA]$^+$ and the off-centering distortion of the inorganic

CuCl$_6$ framework[48]. The decomposition of the polarization indicates that the contribution from the organic cations ($P_{ab, MPA}$ ~ 7.3 μC cm$^{-2}$) is much higher than that arising from the distortion of inorganic framework ($P_{ab, CuCl_4}$ ~ 0.5 μC cm$^{-2}$) (Fig. 4b). The organic component dominates the in-plane polarization ($P_{ab}$), while the inorganic framework component dominates the small out-of-plane polarization ($P_c$).

In addition, we also calculated the spin-polarized electronic band structure and density of states (DOS) of $(R/S)$-(MPA)$_2$CuCl$_4$. As displayed in Fig. 4c, the contribution of the inorganic CuCl$_6$ framework is dominant in both the valence band maximum (VBM) and conduction band minimum (CBM). Here the CBM lies on the $X/Y$ high symmetry $k$-point, while the VBM lies at the $\Gamma$ point, thereby resulting in an in-direct bandgap of ~1.2 eV, which is assigned to the Cu$^{2+}$ $d$-$d$ transitions in near-infrared absorption with a broad band of 700 ~ 1000 nm (Supplementary Fig. 14). The CBM, calculated with and without SOC, exhibits a minimal energy change of ~1.1 meV. It shows two distinct bands separated by an energy difference of ~ 0.26 eV, with the associated charge density localized at the two inequivalent Cu$^{2+}$ sites within the J-T-distorted framework. Besides, as displayed in Supplementary Fig. 15, the combination of the J-T and ferroelectric distortions splits the states of the two Cu$^{2+}$ ions at CBM. This splitting vanishes when the J-T and ferroelectric orders are decoupled; the states become degenerate

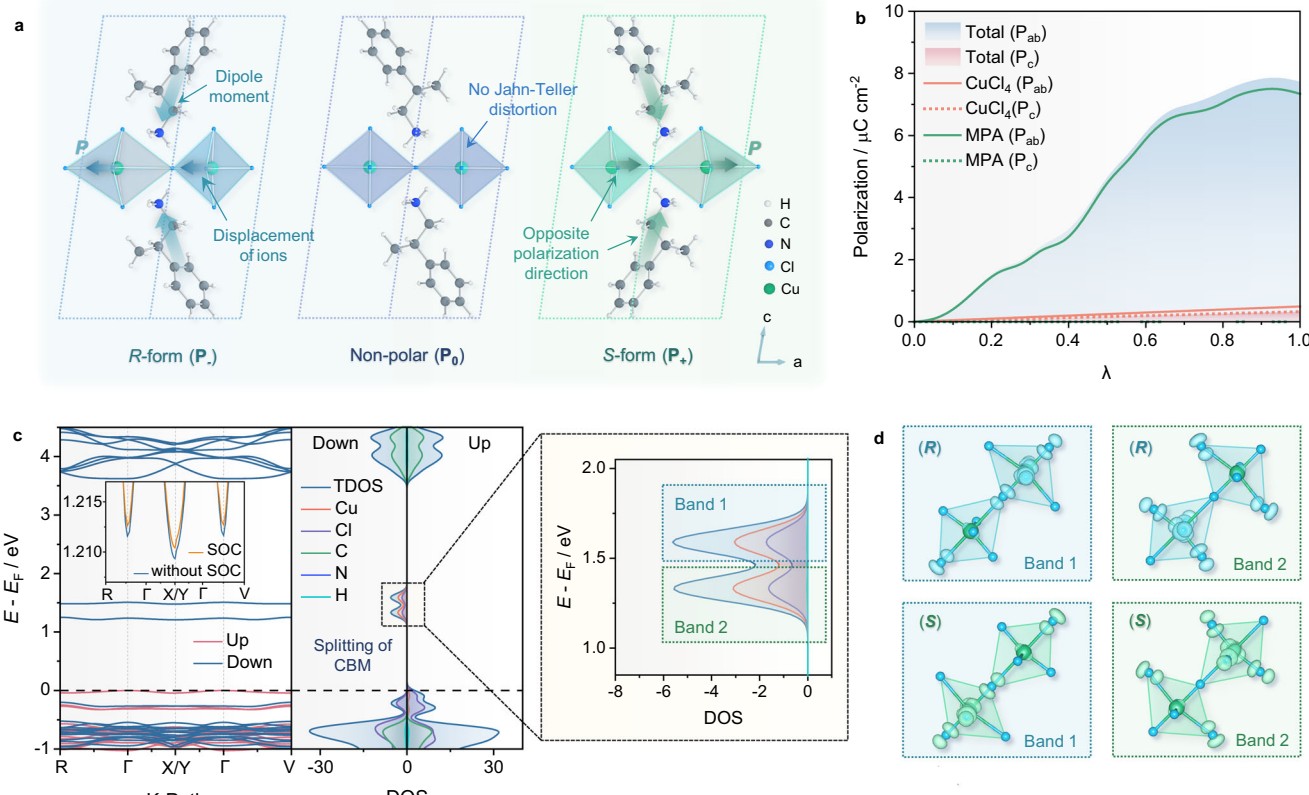

**Fig. 4 | Ferroelectric polarization and electronic structure calculations for (*R/S*)-(MPA)₂CuCl₄.** **a** Crystal structures of polar $P_-$ (in the *R* form) and $P_+$ (in the *S* form) compounds as well as the corresponding non-centrosymmetric non-polar $P_0$ reference. **b** Polarization as a function of $\lambda$ between the non-polar $P_0$ structure ($\lambda = 0$) and the polar $P_+$ structure ($\lambda = 1$). **c** Spin-polarized electronic band structure and density of states for (*R*)-(MPA)₂CuCl₄. The enlarged view shows the splitting of conduction band minimum (CBM) because of J-T distortion and polar displacements. **d** Enlarged image of the electronic doublet states and their respective partial charge densities with a long Cu²⁺-Cl⁻ bond perpendicular and parallel to the ferroelectric distortion, respectively.

when either the J-T distortion is removed while retaining the ferroelectric distortion, or vice versa. The partial charge density plot illustrated in Fig. 4d supports the inequivalence of the two Cu²⁺ sites. Furthermore, the localization of the charge density associated with the two bands switches between the two inequivalent Cu²⁺ sites when the chirality is switched from *R* to *S* or vice versa (see Fig. 4d from top to bottom). This clearly highlights the interplay between crystallographic chirality, J-T distortion, and the underlying electronic properties.

## Simulation of magnetic spin configurations

In order to study the magnetic spin configuration of (*R/S*)-(MPA)₂CuCl₄, we calculated the magnetic exchange coupling parameters ($J_{ij}$), the competing D-M antisymmetric exchange interaction parameters of neighboring Cu²⁺ ions (i.e., the vector $D_i$), and the single ion anisotropy (SIA) parameters in the presence of SOC effect, all of which are tabulated in Supplementary Table 4. The resulting calculated $J_{ij}$ is isotropic, with $J_{ij}^{aa} = J_{ij}^{bb} = J_{ij}^{cc} \sim 5.5$ and 5.7 meV for (*R*)- and (*S*)-(MPA)₂CuCl₄ respectively, suggesting a strong ferromagnetic coupling between magnetic Cu²⁺ ions within the CuCl₆ plane (see Supplementary Note 3 for more computational details). Our experimental results suggest an out-of-plane antiferromagnetic coupling, i.e., A-type antiferromagnet, but to prove that theoretically is challenging due to the long interlayer distance of ~18 Å. The easy-axis of the Cu²⁺ spins were calculated by pointing the spins along in-plane [1 1 0], [$\bar{1}$ 1 0] and [$\bar{1}$ $\bar{1}$ 0] directions as well as the out-of-plane *c*-axis. The corresponding energy variations presented in Supplementary Fig. 16 suggest that the spins prefer aligning collinearly or anti-collinearly to the ferroelectric polar [1 1 0] direction, which exhibits the lowest energy. In addition, the polar crystallographic structures without inversion symmetry result in non-

zero *a*, *b*, and *c*-axis components of the D-M vector as well as a slight *c*-axis canting (as shown in Supplementary Table 4). Notably, when the chirality of crystal structure changes from *R* to *S*, the $D_a$ and $D_b$ components are also interchanged along with a sign change for the $D_c$ component, which suggests chirality-dependent switching of magnetic spin orientation.

## Symmetry mode analysis

Our previous experimental measurements have confirmed the coexistence of ferroelectric, magnetic, and chirality order parameters in (*R/S*)-(MPA)₂CuCl₄. Here, using Landau-type symmetry mode analysis, we investigate the cross-coupling of ferroelectric, magnetic orders with chirality transfer and propose a magnetoelectric coupling mechanism. The symmetry mode analysis involves the calculation of the symmetry-breaking order parameters that arise from the phase transition between the high-symmetry 'parent' and low-symmetry 'child' states with the 'group-subgroup' relationship. The inorganic CuCl₆ framework of (*R*)- and (*S*)-(MPA)₂CuCl₄ possess triclinic *P*1 symmetry, which can be viewed as possessing orthorhombic pseudosymmetry described by 'parent' space group *Cmmm* (#65) (see Supplementary Table 5). Herein, we used irreducible representations (IRREPs) and order parameter directions (OPDs) of space group *Cmmm* to analyze each of the important distortion modes that deform this idealized parent framework structure into the observed triclinic framework structure. These various framework distortions can be described as distortion modes belonging to IRREPs of *Cmmm* at the *Γ* [0,0,0] and *R* [½,½,½] points of the first Brillouin zone (more details can be found in Supplementary Note 4 and Supplementary Note 5).

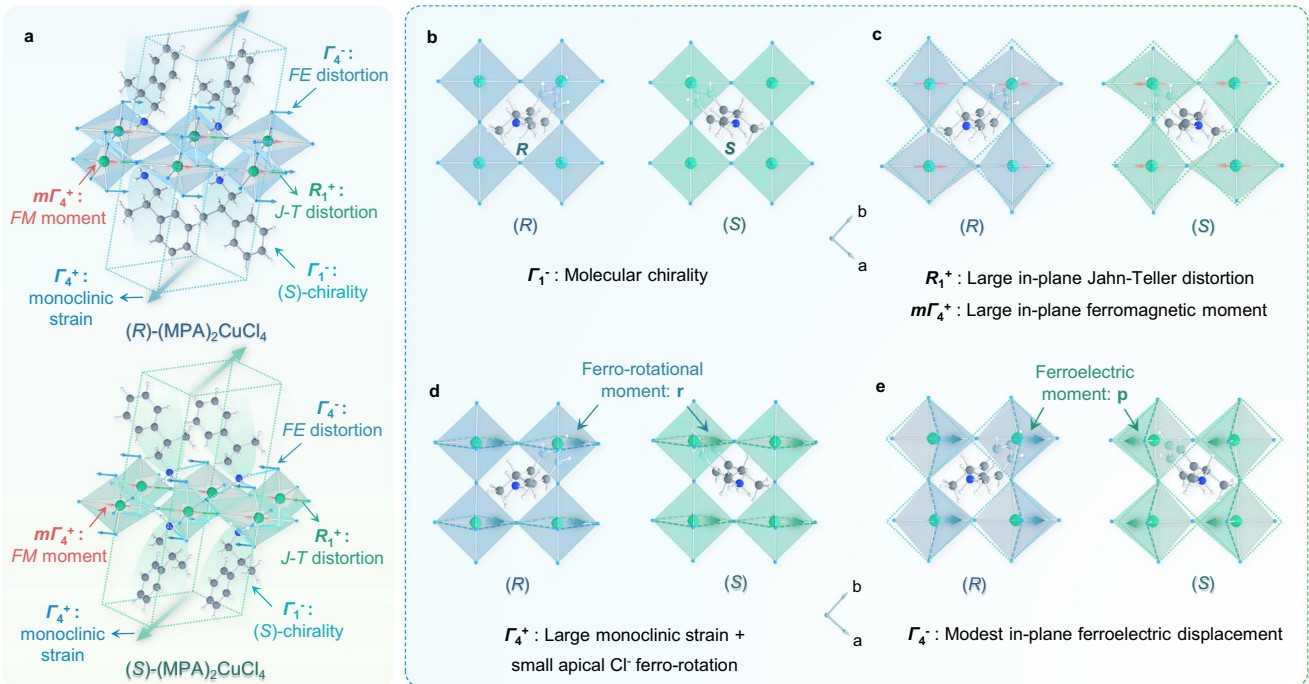

**Fig. 5 | Sketch of perovskite framework showing the primary distortion modes in (R/S)-(MPA)₂CuCl₄. a** Unit cells of $(R/S)$-(MPA)$_2$CuCl$_4$ showing the primary distortion modes of $\Gamma_1^-$, $R_1^+$, $\Gamma_4^+$, $\Gamma_4^-$, and $m\Gamma_4^+$. **b** The mode pattern of non-polar 'parent' *Cmmm* structure showing organic molecular chirality belonging to $\Gamma_1^-$ order parameter. **c** The mode pattern of large in-plane J-T distortion belonging to $R_1^+$ order parameter, and large in-plane ferromagnetic moment belonging to $m\Gamma_4^+$ order parameter. **d** The mode pattern of large monoclinic strain and small apical Cl⁻ ferro-rotation belonging to $\Gamma_4^+$ order parameter. The directions of the axial atom-pair rotation vectors are consistent for $(R)$- and $(S)$-form structures, denoted by ferro-rotational moment **r**. **e** The mode pattern of modest in-plane ferroelectric displacement belonging to $\Gamma_4^-$ order parameter. The directions of the polar ferroelectric displacement vectors are opposite for $(R)$- and $(S)$-form structures, denoted by ferroelectric moment **p**.

As displayed in Fig. 5a and Supplementary Tables 6–8, the IRREP $\Gamma_1^-$ represents molecular chirality and $\Gamma_4^+$ enacts a substantial monoclinic shear strain of the crystal lattice around the parent *b*-axis. $\Gamma_4^+$ also simultaneously contributes compensating atomic displacements that effectively rotate each pair of apical Cl atoms around the parent *b*-axis, which almost reverses the octahedral distortions. The large intra-layer ferromagnetic moments can be described using IRREPs $m\Gamma_4^+$ (parent *b* axis), $m\Gamma_3^+$ (parent *c*-axis), while the small out-of-plane magnetic canting can be described using IRREP $m\Gamma_2^+$ (parent *a*-axis). The large primary J-T distortion belongs to IRREP $R_1^+$ and special OPD $(0; a)$, and is accompanied by two secondary modes of the same wavevector. The imbalanced octahedral scissor distortions are associated with $R_2^+ (0; a)$ and $R_2^- (0; a)$, which describe an antiferro-rotational pattern of apical-Cl pair rotations, and an antiferroelectric pattern of apical-Cl displacements along the in-plane parent *b*-axis, respectively.

As explained in Supplementary Note 6, when considering a chiral-$(R)$ or chiral-$(S)$ ordering of the MPA⁺ molecules compared to a disordered molecule model in a *Cmmm* parent structure, we observe the presence of pre-existing occupational order parameters belonging to $\Gamma_1^-$ and $\Gamma_4^+$. This drives a significant lattice strain and a small compensating ferro-rotation of the apical-Cl atoms belonging to $\Gamma_4^+$ (Fig. 5b, d). The invariant trilinear coupling term in the free energy then gives rise to a ferroelectric moment belonging to $\Gamma_4^-$ (Fig. 5e). While $\Gamma_4^-$ leads to ferroelectric order parameters and a net polarization, $\Gamma_1^-$ and $\Gamma_4^+$ separately result in structures that possess spatial inversion symmetry; therefore, they are achiral modes. This analysis of invariant free-energy polynomials, comprised of order parameters of topological space group *Cmmm*, reveals a trilinear coupling of the $\Gamma_1^-$, $\Gamma_4^+$, and $\Gamma_4^-$. Thus, if any two of them are active in the structure, the free energy can be further lowered by activating the third one as well. Consequently, the pre-existing contributions of $\Gamma_1^-$ and $\Gamma_4^+$ to the ordered chiral molecular configuration of $(R)$- or $(S)$-(MPA)$_2$CuCl$_4$ naturally couple to

ferroelectric displacements $\Gamma_4^-$ of the MPA⁺ cations, which can reasonably be characterized as a hybrid-improper ferroelectric mechanism.

It is important to note that a mirror operation in the triclinic [1,1,0] direction reverses the polarity of the ferroelectric displacement vectors (total ferroelectric moment **p**), but preserves the direction of the axial atom-pair rotation vectors (ferro-rotational moment **r**) (as depicted in Figs. 5d, e). By combining these polar and axial moments, we define $\xi = \mathbf{p} \cdot \mathbf{r}$ as a pseudo-scalar property of the framework. Its sign differentiates between opposite framework chiralities, with sign($\xi$) = +1 and −1 indicating the molecular chirality of $(R)$- and $(S)$-(MPA)$_2$CuCl$_4$, or vice versa. Therefore, we propose a symmetry-mode analysis that includes crystallographic chirality as an order parameter, allowing for the formal definition of chirality transfer through a group theoretical approach. Our study of the chiral multiferroic HOIPs reveals: (i) a trilinear coupling amongst chiral, ferroelectric, and ferro-rotational order parameters, (ii) a hybrid-improper mechanism of ferroelectric polarization, (iii) a simple pseudo-scalar descriptor of framework chirality, and (iv) the hybrid-improper character of the chirality transfer from the organic MPA⁺ subsystem to the inorganic framework. These properties may be probably found in similar hybrid systems where the chirality transfer is active.

Additionally, the trilinear coupling of $\Gamma_4^-$, $\Gamma_4^+$, and $\Gamma_1^-$ is accompanied by a J-T-type antiferrodistortion in $(R)$- and $(S)$-chiral HOIPs (Fig. 5c), denoted as $R_1^+$, which is essential to the intra-layer ferromagnetic orders ($m\Gamma_4^+$, $m\Gamma_3^+$ and $m\Gamma_2^+$). The interplay between J-T distortion and the SOC effect is directly linked to the orbital magnetic moments of HOIPs, where switching the orientation of the J-T distortion interconverts the orthogonality of magnetic Cu²⁺ 3*d* orbital ordering. This observation is consistent with the chirality-induced interchange of D-M vectors and the mirrored charge density patterns confirmed by theoretical calculations. This presents a possible

scenario where the transfer of chirality from the organic molecules to the inorganic octahedra in HOIPs facilitates a coherent coupling between the in-plane ferroelectric and ferromagnetic orders (Supplementary Note 7 and Supplementary Note 8 for more details), which has yet to be experimentally proven.

## Discussion

In this study, we have presented experimental evidence of chiral multiferroic properties in layered copper-based HOIPs $(R/S)$-$(MPA)_2CuCl_4$. These materials exhibit in-plane ferroelectric behavior, A-type antiferromagnetic configuration, and chirality-dependent MCD characters. Through the application of Landau symmetry-mode analysis, we have identified a trilinear coupling mechanism that links molecular chirality with the non-chiral ferroelectric and ferrorotational moments of the inorganic framework. This hybrid-improper mechanism enables the transfer of chirality from the organic molecules to the inorganic framework. The extent of chirality transfer and the differentiation between the $(R)$- and $(S)$-enantiomers can be quantified by the pseudo-scalar quantity $\xi = \mathbf{p} \cdot \mathbf{r}$.

Furthermore, the symmetry-allowed couplings between framework chirality, J-T antiferro-distortion, D-M vectors, and intra-layer ferroelectric and ferromagnetic orders introduce the alluring possibility of observing novel chirality-assisted magnetoelectric phenomena. By leveraging the interplay between chirality and multiple ferroic orders, we can unlock the avenues for designing and manipulating materials with enhanced properties and tailored functionalities in the field of quantum technologies.

## Methods

### Materials

Anhydrous copper chloride ($CuCl_2$, ≥99%), $(R)$-(+)-$\beta$-Methylphenethylamine ($R$-MPA, 99%), $(S)$-$\beta$-Methylphenethylamine ($S$-MPA, 99%) and $\beta$-Methylphenethylamine ($rac$-MPA, 99%) were purchased from Sigma-Aldrich. Methanol (≥99.8 %) and hydrochloric acid (HCl, 37% in $H_2O$) were purchased from VWR chemicals. All chemicals were used as received without further purification.

### Synthesis of perovskite single crystals

$(R/S)$-$(MPA)_2CuCl_4$ were synthesized by slowly evaporating the stoichiometric amounts of organic ligands and $CuCl_2$ in the atmospheric environment. Initially, 1 mmol of $(R/S)$-MPA was neutralized using 1 mmol of HCl (37% in $H_2O$, VWR chemicals) within a 20-mL glass container. Subsequently, 0.5 mmol of $CuCl_2$ was introduced to the mixture, followed by the addition of 15 mL of a solvent blend (methanol: isopropanol = 1:1) to ensure complete dissolution through ultrasonic agitation. The solution was then set aside in a controlled environment to prevent external vibrations. After one month, solvent evaporation under ambient conditions yielded centimeter-sized single crystals. The racemic $(MPA)_2CuCl_4$ was prepared using a similar synthetic method but results in the microcrystalline-form samples.

### Structural and optical characterization

Single-crystal X-ray diffraction (XRD) experiments were carried out utilizing a Bruker D8 Venture X-ray single-crystal diffractometer, employing MoKα radiation ($\lambda = 0.71073$), and conducted at temperatures of 100 K, 150 K, 200 K, and 300 K. The processing of single-crystal XRD data, including integration, scaling, and absorption correction, was executed using APEX 3 (version v2019.11-0). Initial models were deduced employing an intrinsic phasing method through the ShelXT program[49], followed by subsequent structural refinements using the OLEX2 program package. The refinement process employed a graphical interface with full-matrix least-squares on $F^{2,50}$. The corresponding CIF files have been uploaded to the Cambridge Crystallographic Data Center (CCDC-2356454-2356461) and are available for free download. Powder XRD data were obtained employing a Bruker

XRD analyzer D2 phaser with Cu Kα radiation at room temperature. UV optical absorption spectra were recorded using a Shimadzu UV-3600 UV-visible-near-infrared spectrometer. MCD spectroscopy was conducted by fixing the spin-coated perovskite polycrystalline films within a Jasco-1500 CD spectrometer, which was equipped with an adjustable PM-491 permanent magnet accessory capable of generating maximum fields of ±1.6 T.

### Magnetic characterization

The magnetic properties were characterized using a Quantum Design 7 Tesla superconducting quantum interference device (SQUID) magnetometer. The perovskite single crystal samples were fixed onto a silicon substrate with Kapton tape and oriented in different crystallographic directions relative to the applied magnetic field. The temperature-dependent magnetization was measured from 3 K to 15 K under selected field values. The field dependences of the magnetization were measured up to ±5 kOe. Broadband ferromagnetic resonance (FMR) measurements were performed using a Quantum Design CryoFMR setup. The single-crystal samples were placed on a coplanar waveguide (CPW) positioned between two Helmholtz coils within a physical property measurement system (PPMS). As the resonance condition is fulfilled, the sample absorbs energy from the CPW, which manifests as a decrease in transmitted energy.

### Ferroelectric characterization

Polarization versus voltage ($P$-$V$) hysteresis loop measurements were carried out by using a commercial Precision Multiferroic II Ferroelectric Tester with a double-wave method. High-quality perovskite single crystals were contacted with silver paints along the polar axis to measure the ferroelectric hysteresis loops. Piezoelectric force microscopy (PFM) characterization was performed with a commercial atomic force microscope (AFM) (Bruker Dimension FastScan) in a $N_2$ glovebox. The PFM samples were prepared by dissolving the perovskite single crystals in methanol with a concentration of -10 mg/mL and growing on Si wafers with an annealing temperature of 70 °C. The SCM-PIT-V2 Pt/Ir coating probe was used to collect the PFM phase and amplitude signals with a drive frequency of 295 kHz and drive amplitude of 6500 mV.

### Theoretical calculations

Density-functional-theory calculations were conducted using the Vienna ab initio simulation package within the projector-augmented-waves scheme[51–53]. The Perdew-Burke-Ernzerh function was employed to treat correlation and exchange[54]. The $k$-point mesh was used as $6 \times 6 \times 4$ and the cutoff energy was set at 520 eV. All the structures underwent full relaxation until the forces on all atoms reached values smaller than 0.001 eV Å$^{-1}$. For the $Cu^{2+}$ $3d$ states, we chose an effective Hubbard correction $U - J_H = 6.0$ eV. The SOC was included to calculate the electronic and magnetic properties. The magnetic exchange coupling parameters ($J_{ij}$), the D-M antisymmetric exchange interaction parameters $D_i$, and the SIA parameters were calculated by following the procedures shown in Supplementary Notes[55,56].

## Data availability

All data needed to evaluate the conclusions in the study are present in the paper and/or the Supplementary Information. All data that support the findings within this paper are available from the corresponding authors upon request.

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

## Acknowledgements

K.P.L. acknowledges Singapore's Ministry of Education Tier 2 grant MOE2019-T2-1-037 and Singapore's National Research Foundation grant NRF CRP22-2019-0006. R.M. acknowledges the Ministry of Education for supporting this work (Grant no. A-0004212-00-00 and A-8000462-00-00). A.S. acknowledges the support from the Italian Ministry of Research under the PRIN 2022 Grant No 2022F2K7J5 with the title "Two-dimensional chiral hybrid organic–inorganic perovskites for chiroptoelectronics" PE 3 funded by PNRR Mission 4 Istruzione e Ricerca - Component C2 - Investimento 1.1, Fondo per il Programma Nazionale di Ricerca e Progetti di Rilevante Interesse Nazionale PRIN 2022 – CUP B53D23004130006. A.S. acknowledges P. Barone, K. Yananose, D. Sangalli, and E. Molteni for the useful discussions. A.S., M.J.S., and S.G. acknowledge CINECA for providing the computational facility. K.W. acknowledges the financial support from the Tangshan Science and Technology Bureau (23130226E). M.J.S. acknowledges HPC-Europa3 H2020 Grant Agreement 730897 (application code HPC17TYDIN) for supporting the visiting period at CNR-SPIN c/o Department of Physical and Chemical Science at University of L'Aquila (Italy). M.J.S. acknowledges the kind and warm hospitality during the visiting period from January to April 2022 at CNR-SPIN in L'Aquila. M.J.S. and S.G. acknowledge the DST-NSM National Supercomputing Mission (DST/NSM/R&D_HPC_Application/2021/34) for funding. F.C. is indebted to Romanian Research Ministry for support by CNCS-UEFISCDI, project number PN-III-P4-PCE-2021-1881.

## Author contributions

H.Z. synthesized the perovskites and performed the ferroelectric characterization. A.G. performed the ferromagnetic resonance measurements. M.J.S. performed the theoretical calculations. Q.Z. performed the SQUID measurements. J.C. supervised the SQUID measurements. W.P.D.W and Z.W. analyzed the single-crystal XRD data. R.Z. calculated the electronic structures. F.C. performed the theoretical analysis of molecular dipoles and spatial arrangements. S.G., A.S., and B.J.C. supervised the theoretical analysis and calculations. B.J.C. carried out the symmetry-mode analysis. K.W. performed the MCD measurements. R.M. supervised the ferromagnetic resonance characterization. H.Z. and M.J.S. wrote the manuscript under the guidance of K.P.L., A.S., and B.J.C.

## Competing interests

The authors declare no competing interests.
