## [Peer Review File · Nature Communications]

Chiral Multiferroicity in Two-Dimensional Hybrid Organic-Inorganic PerovskitesEditorial Note: Parts of this Peer Review File have been redacted as indicated to remove third-party material where no permission to publish could be obtained.

REVIEWER COMMENTS

Reviewer #1 (Remarks to the Author):

This paper by Haining Zheng et al mainly focuses on the investigation of the multiferroic properties of 2D HOIPs (R/S)-(MPA)₂CuCl₄. They discussed the cross coupling of chirality and ferroic orders and proposed a new hybrid improper chirality transfer mechanism. This hybrid [(R/S)-(MPA)₂CuCl₄] was systematically analyzed through various characterizations, including single-crystal structure analysis, ferromagnetic and ferroelectric properties, P-V and M-H hysteresis loops, ferroelectric domain, and theoretical calculations. The authors present an excellent research work on developing multiferroic materials.

The manuscript is well written and suitable for publication in Nature Communications.

However, I have several suggestions, and some critical questions must be addressed before its publication.

1, The 2D HOIPs (R/S)-(MPA)₂CuCl₄ have been previously reported by Bing Sun et al. (Chem. Mater. 2020, 32, 8914). The magnetic properties, including M-H hysteresis loops, presented in Sun's work (Figure 3) are similar to those in this study (Figure 2). This similarity raises concerns regarding the novelty of the magnetic aspects of this research. To address this issue, I suggest that the authors should discuss more of Sun's work and highlight the differences and contributions of their own study in the context of the magnetic properties.

2, The authors report that [(R/S)-(MPA)₂CuCl₄] crystallizes in the P1 space group, with data collected at 100K presented in Table S1. However, Sun et al. reported that [(R/S)-(MPA)₂CuCl₄] crystallizes in the C2 space group at 150K according to their CIF files. This discrepancy suggests the possibility of a temperature-dependent structural phase transition. In my opinion, Supplementary Fig. 8 is insufficient to conclusively determine that the ferroelectric Curie temperature of [(R/S)-(MPA)₂CuCl₄] is higher than its melting point. I recommend that the authors reconduct DSC measurements down to lower temperatures to check if there is a phase transition occurred at low temperature.

3. The authors claim that the chirality of the metal-halide framework is induced by the organic cations, [(R/S)-MPA]⁺. This leads me to question how the racemic cation influences the chirality of the inorganic metal-halide framework. I suggest that the authors present, compare, and discuss the crystal structure of the racemic [(MPA)₂CuCl₄] to provide further insights into this aspect.

Reviewer #2 (Remarks to the Author):

Comments:

In this article, the authors rediscovered the ferroelectricity present in the previously reported compound described in Chem. Mater. 2020, 32, 20, 8914–8920, and proposed a new mechanism for chiral transfer. Through Landau symmetry mode analysis, the authors analyzed the cross-coupling of chirality with ferroic orders and proposed a new hybrid

improper chirality transfer mechanism, for which the coupling of two non-chiral distortions enables the transfer of chirality from the organic cations to the framework and proposed a new pseudo-scalar quantity, $\xi = \mathbf{p} \cdot \mathbf{r}$ to describe coupling. Additionally, the authors supplemented the study with MCD (Magnetic circular dichroism) measurements and other relevant tests on the compound.

- 1) Can ferroelectric and magnetic properties be coupled in the compound? If conditions permit, can the flipping of ferroelectric domains be controlled by a magnetic field Science 367,671-676(2020)?
- 2) As mentioned by the authors in Chem. Mater. 2020, 32, 20, 8914–8920, the presence of magneto-chiral dichroism (MChD) in this structure has already been reported, indicating the existence of MCD in the compound. The authors supplemented the MCD spectra and conducted comparisons. How does this gmcd value compare to the devices already in use?
- 3) The authors mentioned that 'The effect originates from the fine-tuning of the chiral-dependent exciton energies by the field-induced Zeeman splitting', where the applied magnetic field induces perturbations in the states of transition dipoles (ground, excited, and other energy states), resulting in a universally present and temperature-independent MCD. How did the authors eliminate this interference?
- 4) The author mentions that between the layers of the compound, there is a ferromagnetic arrangement and a DM interaction in the helical structure. However, from the structure, it can be observed that the Cu-Cl-Cu exhibits an approximately 180° antiferromagnetic arrangement, which contradicts the Goodenough-Kanamori rules.
- 5) In line 218, the authors mentioned the ferroelectric-to-paramagnetic phase transition. Considering the context, should it be interpreted as ferroelectric-to-paraelectric phase transition?
- 6) In Table 1, P1, as a non-centrosymmetric polar space group, should provide the Flack parameter.

Point-To-Point Response to Reviewers' Comments

The reviewers' comments are marked in **black**, and the author's responses are in **blue**. The major changes in manuscript and Supplementary Information (SI) have been highlighted by **yellow**.

Reviewer: 1

Comments:

This paper by Haining Zheng et al mainly focuses on the investigation of the multiferroic properties of 2D HOIPs (*R/S*)-(MPA)₂CuCl₄. They discussed the cross coupling of chirality and ferroic orders and proposed a new hybrid improper chirality transfer mechanism. This hybrid [(*R/S*)-(MPA)₂CuCl₄] was systematically analyzed through various characterizations, including single-crystal structure analysis, ferromagnetic and ferroelectric properties, *P-V* and *M-H* hysteresis loops, ferroelectric domain, and theoretical calculations. The authors present an excellent research work on developing multiferroic materials.

The manuscript is well written and suitable for publication in Nature Communications. However, I have several suggestions, and some critical questions must be addressed before its publication.

Response: We thank the reviewer for the high evaluation and recognition of the quality of this work.

1. The 2D HOIPs (*R/S*)-(MPA)₂CuCl₄ have been previously reported by Bing Sun et al. (Chem. Mater. 2020, 32, 8914). The magnetic properties, including *M-H* hysteresis loops, presented in Sun's work (Figure 3) are similar to those in this study (Figure 2). This similarity raises concerns regarding the novelty of the magnetic aspects of this research. To address this issue, I suggest that the authors should discuss more of Sun's work and highlight the differences and contributions of their own study in the context of the magnetic properties.

Response: We thank the reviewer for the constructive comments. In Sun's work (Sun *et al.* Chem. Mater., **32**, 8914 (2020)), which is highly instructive and provides significant inspiration for our study, they reported the first example of layered hybrid perovskite chiral ferromagnets by

incorporating chiral cations [(*R/S*)-MPA]⁺ into the inorganic CuCl₆ octahedral frame. They found that these perovskites show typical ferromagnetic hysteresis loops with T_c of 6 K and high saturated magnetization of 12.5 emu g⁻¹. The ferromagnetic coupling arises from the orthogonal arrangement of unoccupied Cu $d(x^2-y^2)$ orbitals induced by Jahn-Teller distortion. However, the samples used for magnetization characterization were **in powder form**, which did not permit the exploration of the **magnetic anisotropy in this compound**. That work also lacked a detailed **theoretical simulation of magnetic spin configuration as well as the estimation of the ferroelectric polarization** and discussions of the correlations among magnetic spin configuration, chirality and ferroelectric order parameters.

Compared to Sun's work, the key novelty of this work regarding the magnetic properties is summarized as follows:

- (1) We synthesized the centimeter-sized high-quality **single crystals** and characterized the magnetic properties along the crystallographic ***a***, ***b***, and ***c***-axis, respectively. We found that (*R/S*)-(MPA)₂CuCl₄ displays strong magnetic anisotropy which shows soft **ferromagnetic behavior** within the CuCl₆ plane but is **antiferromagnetically coupled** along interlayer direction. The out-of-plane antiferromagnetic properties can be readily transferred to ferromagnetic coupling when the magnetic fields exceed 4000 Oe; that may be why Sun *et al.* did not observe this phenomenon using powder-form samples. This indicates that this compound is a typical **A-type antiferromagnet** rather than a pure ferromagnet.
- (2) We also performed the **ferromagnetic resonance (FMR)** to further examine the magnetic anisotropy. We found that the FMR spectra present inverse variation of resonance peaks along intralayer and interlayer directions, which can be attributed to the **strong magnetocrystalline anisotropy**, and the magnetic easy axis lies in the *ab* plane. The linewidth of FMR peaks also indicates that (*R/S*)-(MPA)₂CuCl₄ present long spin lifetimes τ of 2.702 ± 0.327 ns (intralayer) and 1.590 ± 0.144 ns (interlayer), which are much longer than typical Pb²⁺- and Sn²⁺-based perovskites (Wang *et al. J. Mater. Chem. C* **6**, 2989-2995 (2018); Liang *et al. ACS Energy Lett.* **6**, 1670-1676 (2021)).

(3) We further performed theoretical calculations to simulate the **magnetic spin configurations** by calculating the exchange coupling parameters (J_{ij}), D-M exchange parameters (D_{ij}), and Single Ion Anisotropy (SIA) parameters. We found that the A-type antiferromagnetic feature originates from the **geometrical anisotropy** of stratified Ruddlesden–Popper-type configuration. The in-plane D_{ij} components are also interchanged when switching the crystallographic chirality, suggesting the chirality-dependent magnetic spin orientation.

To clarify this point, we have added some discussion in the manuscript as follows:

“Magnetic anisotropy. A recent report by Sun et al. revealed that the powder-form crystals of (R/S) -(MPA)₂CuCl₄ exhibit soft ferromagnetic properties, with a Curie temperature T_c of approximately 6 K¹⁵. Herein, we focus on high-quality single-crystal samples and examine their magnetic anisotropy by characterizing the magnetic hysteresis loops along the a , b , and c -axis directions, respectively.....

.....By contrast, a clear antiferromagnetic transition occurs at $T = 4$ K when the magnetic field is applied perpendicular to the CuCl₆ plane ($H_{\text{ext}} \parallel c$ -axis); it can be easily converted to ferromagnetic ordering under higher external magnetic fields (> 1 kOe) (**Figs. 2b** and **f**). In contrast to the ferromagnetic behavior observed in Sun’s work, our findings indicate that (R/S) -(MPA)₂CuCl₄ display intra-layer ferromagnetic and long-range inter-layer antiferromagnetic properties, i.e. a typical A-type antiferromagnetic pattern. Such a pattern is associated with the geometrical anisotropy of stratified Ruddlesden–Popper-type configuration, where the magnetic CuCl₆ frame is separated by non-magnetic (R/S) -(MPA)⁺ cations along the c -axis²⁴. The intra-layer ferromagnetic behavior is also in agreement with the Goodenough–Kanamori rules. This is a consequence of the orbital orthogonality of the magnetic Cu²⁺ 3d orbitals^{15,25}, induced by the alternate orientation (mutually perpendicular) of the J-T elongation axes within the 2D framework^{22,24}.

REFERENCE

15 Sun, B. *et al.* Two-dimensional perovskite chiral ferromagnets. *Chem. Mater.* **32**, 8914-8920 (2020).

22 Stroppa, A., Barone, P., Jain, P., Perez-Mato, J. M. & Picozzi, S. Hybrid improper ferroelectricity in a multiferroic and magnetoelectric metal-organic framework. *Adv. Mater.* **25**, 2284-2290 (2013).

24 Comstock, A. H. *et al.* Hybrid magnonics in hybrid perovskite antiferromagnets. *Nat. Commun.* **14**, 1834 (2023).

25 Taniguchi, K. *et al.* Magneto-electric directional anisotropy in polar soft ferromagnets of two-dimensional organic–inorganic hybrid perovskites. *Angew. Chem. Int. Ed.* **60**, 14350-14354 (2021).”

2. The authors report that [(*R/S*)-(MPA)₂CuCl₄] crystallizes in the *P1* space group, with data collected at 100 K presented in Table S1. However, Sun *et al.* reported that [(*R/S*)-(MPA)₂CuCl₄] crystallizes in the *C2* space group at 150 K according to their CIF files. This discrepancy suggests the possibility of a temperature-dependent structural phase transition. In my opinion, Supplementary Fig. 8 is insufficient to conclusively determine that the ferroelectric Curie temperature of [(*R/S*)-(MPA)₂CuCl₄] is higher than its melting point. I recommend that the authors reconduct DSC measurements down to lower temperatures to check if there is a phase transition occurred at low temperature.

Response: We acknowledge the reviewer for these useful comments. After carefully analyzing the crystal structures and crystallographic symmetry relationships, we found that *P1* is a more correct space group for (*R/S*)-(MPA)₂CuCl₄. As illustrated in **Figure R1 (a)**, (*R/S*)-(MPA)₂CuCl₄ have disordered Cl atoms (Cl₂, Cl₃, Cl₄) within the CuCl₆ octahedral plane. If we index with *C2* space group, these disordered Cl atoms are symmetrically equivalent (especially Cl₂), which is inconsistent with the well-known Jahn-Teller antiferrodistortive arrangement of the Cu²⁺-based octahedral perovskite structure. If we index with space group *P1*, these Cl atoms are **inequivalent**, with site occupancy factors (SOF) of 0.915 and 0.085, respectively. This aligns nicely with the presence of a large Jahn-Teller distortion having alternately elongated and compressed Cu-Cl bonds in plane. The *P1* space group was also reported in another paper investigating the magneto-electric directional anisotropy in (*R/S*)-(MPA)₂CuCl₄ (**Figure R1 (b)**) (Taniguchi, K. *et al.* *Angew. Chem. Int. Ed.* **60**, 14350-14354 (2021)).

In order to examine possible phase transitions, we have carried out DSC measurements to lower

temperatures. However, owing to the limitations of our chiller system, our DSC instrument can only reach a minimum temperature of 220 K. Supplementary Fig. 9 (b) presents the new DSC results from 220 K to 525 K. It shows that there is no distinct heat flow peak until decomposition. At lower temperatures, we supplemented the single-crystal X-ray diffraction (XRD) measurements to determine if there is any change in space groups. As demonstrated in Supplementary Tables 1-2, (*R/S*)-(MPA)₂CuCl₄ crystallizes into space group *P1* within the temperature range from 100 K to 300 K, thus providing additional evidence that they **do not undergo a phase transition** at lower temperatures.

[Redacted]

Figure R1. (a) Schematic diagrams showing the disordered Cl atoms within the CuCl₆ plane for space groups *C2* and *P1*, respectively. (b) Crystal structure of (*R/S*)-(MPA)₂CuCl₄ with space group *P1* reported by Taniguchi *et al.* (Taniguchi, K. *et al. Angew. Chem. Int. Ed.* **60**, 14350-14354 (2021))

To further clarify this point, we have added the new DSC results (from 220 K to 526 K) and the single-crystal XRD data (100K, 150 K, 200 K, and 300 K) in the SI as follows:

Supplementary Fig. 9. (a) Thermogravimetric analysis (TGA) curves of (R/S) -(MPA)₂CuCl₄. (b) Differential scanning calorimetry (DSC) curves of (R/S) -(MPA)₂CuCl₄.

(R) -(MPA) ₂ CuCl ₄				
Empirical formula	C ₁₈ H ₂₈ Cl ₄ CuN ₂	C ₁₈ H ₂₈ Cl ₄ CuN ₂	C ₁₈ H ₂₈ Cl ₄ CuN ₂	C ₁₈ H ₂₈ Cl ₄ CuN ₂
Formula weight	477.76 g/mol	477.76 g/mol	477.76 g/mol	477.76 g/mol
Temperature/K	100.00 K	150.00 K	200.00 K	300.00 K
Crystal system	triclinic	triclinic	triclinic	triclinic
Space group	P 1	P 1	P 1	P 1
a /Å	7.5711(3) Å	7.5916(4) Å	7.6137(3) Å	7.6655(3) Å
b /Å	7.5732(3) Å	7.5949(3) Å	7.6180(2) Å	7.6684(3) Å
c /Å	17.8700(7) Å	17.8982(9) Å	17.9298(6) Å	17.9969(7) Å
α /°	80.8830(10)°	80.926(2)°	80.9850(10)°	81.1260(10)°
β /°	86.802(2)°	86.753(2)°	86.6570(10)°	86.4910(10)°
γ /°	89.4090(10)°	89.3770(10)°	89.3960(10)°	89.4380(10)°
Volume/Å ³	1010.10(7) Å ³	1017.41(8) Å ³	1025.35(6) Å ³	1043.27(7) Å ³
Z	2	2	2	2
Density (calculated)	1.571 g/cm ³	1.560 g/cm ³	1.547 g/cm ³	1.521 g/cm ³
Absorption coefficient	1.615 mm ⁻¹	1.603 mm ⁻¹	1.591 mm ⁻¹	1.563 mm ⁻¹
F(000)	494.0	494.0	494.0	494.0

2 θ range for data collection/ $^{\circ}$	4.624 to 61.066	4.616 to 61.024	4.608 to 61.122	4.59 to 61.06
Reflections collected	56229	54074	55860	54316
Data/restraints/parameters	12271/95/528	12336/95/528	12490/95/528	12700/95/528
Goodness-of-fit on F ²	1.043	1.021	1.033	1.030
Final R indexes [I> \geq 2 σ (I)]	R ₁ = 0.0270, wR ₂ = 0.0594	R ₁ = 0.0258, wR ₂ = 0.0594	R ₁ = 0.0294, wR ₂ = 0.0628	R ₁ = 0.0355, wR ₂ = 0.0747
Flack parameter	0.001(4)	-0.003(3)	-0.003(4)	-0.002(5)

Supplementary Table 1. Single-crystal XRD results for (R)-(MPA)₂CuCl₄.

(S)-(MPA) ₂ CuCl ₄				
Empirical formula	C ₁₈ H ₂₈ Cl ₄ CuN ₂	C ₁₈ H ₂₈ Cl ₄ CuN ₂	C ₁₈ H ₂₈ Cl ₄ CuN ₂	C ₁₈ H ₂₈ Cl ₄ CuN ₂
Formula weight	477.76 g/mol	477.76 g/mol	477.76 g/mol	477.76 g/mol
Temperature/K	100.00 K	150.00 K	200.00 K	300.00 K
Crystal system	triclinic	triclinic	triclinic	triclinic
Space group	P 1	P 1	P 1	P 1
a/Å	7.5876(3) Å	7.5917(3) Å	7.6164(2) Å	7.6652(3) Å
b/Å	7.5928(3) Å	7.5934(3) Å	7.6171(3) Å	7.6741(2) Å
c/Å	17.9130(7) Å	17.8945(7) Å	17.9279(6) Å	18.0025(7) Å
α / $^{\circ}$	80.8960(10) $^{\circ}$	80.9730(10) $^{\circ}$	81.0150(10) $^{\circ}$	81.1800(10) $^{\circ}$
β / $^{\circ}$	86.8300(10) $^{\circ}$	86.7510(10) $^{\circ}$	86.6890(10) $^{\circ}$	86.532(2) $^{\circ}$
γ / $^{\circ}$	89.3750(10) $^{\circ}$	89.3910(10) $^{\circ}$	89.3970(10) $^{\circ}$	89.4260(10) $^{\circ}$
Volume/Å ³	1017.43(7) Å ³	1017.15(7) Å ³	1025.61(6) Å ³	1044.53(6) Å ³
Z	1	2	2	2
Density (calculated)	1.560 g/cm ³	1.560 g/cm ³	1.547 g/cm ³	1.519 g/cm ³
Absorption coefficient	1.603 mm ⁻¹	1.604 mm ⁻¹	1.590 mm ⁻¹	1.562 mm ⁻¹
F(000)	494.0	494.0	494.0	494.0
2 θ range for data collection/ $^{\circ}$	4.612 to 60.974 $^{\circ}$	4.616 to 61.012 $^{\circ}$	4.608 to 61.008 $^{\circ}$	4.588 to 61.004 $^{\circ}$
Reflections collected	66898	74460	77766	59409

Data/restraints/parameters	12244/95/504	12367/95/504	12466/94/504	12714/94/504
Goodness-of-fit on F^2	1.042	1.024	1.046	1.037
Final R indexes [$I \geq 2\sigma$ (I)]	$R_1 = 0.0274,$ $wR_2 = 0.0595$	$R_1 = 0.0308,$ $wR_2 = 0.0644$	$R_1 = 0.0298,$ $wR_2 = 0.0614$	$R_1 = 0.0384,$ $wR_2 = 0.0772$
Flack parameter	-0.010(3)	-0.004(4)	-0.003(4)	-0.003(6)

Supplementary Table 2. Single-crystal XRD results for (S) -(MPA)₂CuCl₄.”

3. The authors claim that the chirality of the metal-halide framework is induced by the organic cations, [(*R/S*)-MPA]⁺. This leads me to question how the racemic cation influences the chirality of the inorganic metal-halide framework. I suggest that the authors present, compare, and discuss the crystal structure of the racemic [(MPA)₂CuCl₄] to provide further insights into this aspect.

Response: Thank you very much for this nice suggestion. We agree that the discussion of crystal structure in racemic [(MPA)₂CuCl₄] will provide further insights into the chiral transfer between organic cations and inorganic metal-halide framework. However, after multiple attempts, we found that the racemic structure presents challenges for the production of high-quality single crystals comparable to their chiral counterparts. They consistently grow into powder-form polycrystals that are not suitable for single-crystal XRD measurements. This has also been observed in previous reports on this compound (MPA)₂CuCl₄, such as Sun *et al. Chem. Mater.*, **32**, 8914 (2020) (mentioned by the reviewer) and Taniguchi *et al. Angew. Chem. Int. Ed.* **60**, 14350 (2021), which also **did not** provide racemic crystal structures. This is probably because the racemic (MPA)⁺ cations always lead to a racemically balanced polycrystalline mixture of very small but enantio-pure crystallites. For example, in Sun *et al.*'s paper, they mentioned that:

“.....By applying the racemic *rac*-MPEA·HCl in the synthesis, we also obtained a racemic compound (*rac*-MPEA)₂CuCl₄. Although the crystal quality was not good enough for single-crystal structural analysis, we confirmed that (*rac*-MPEA)₂CuCl₄ crystallizes in a different structure from those of (*R*-MPEA)₂CuCl₄ and (*S*-MPEA)₂CuCl₄ by comparing PXRD patterns (Sun *et al. Chem. Mater.*, **32**, 8914 (2020)).”

Nevertheless, we can easily construct a **racemic non-polar reference structure** by replacing half of the molecules with the other enantiomer partner and by enforcing spatial inversion symmetry on the CuCl₄ framework. Both (*R*)- and (*S*)-(MPA)⁺ molecules are simultaneously present in pairs within the unit cell and are **related by inversion symmetry**. This corresponds to imposing the inversion symmetry in the unit cell. Furthermore, as discussed in the manuscript, the degree of chirality transfer can be described by the pseudo-scalar quantity $\xi = \mathbf{p} \cdot \mathbf{r}$ (where \mathbf{p} is the ferroelectric displacement vector and \mathbf{r} is the ferro-rotational vector), which differentiates the (*R*)- and (*S*)-enantiomers by its sign. In the racemic structure, the presence of inversion symmetry constrains the ferroelectric polarization to vanish; consequently, the ferroelectric displacement vector also vanishes, resulting in the chirality transfer scalar quantity $\xi = \mathbf{0}$. On the other hand, due to the presence of opposite enantiomers on either side of the framework, the non-symmetric displacement of the apical Cl atoms, caused by the interaction between the framework and the molecule via the C-H-Cl bond, also vanishes. Thus, in our considered racemic structure, the chirality transfer between organic and inorganic frameworks is zero, as expected by symmetry.

To further clarify this point, we have added more discussions in the manuscript and **SI** section 4.3 as follows:

“In order to gain insights into the origin of ferroelectric polarization, we have constructed a chirality-preserving transition path by introducing a non-centrosymmetric non-polar reference structure. **This structure is constructed by in-plane rotating half of the organic molecules by 180° to compensate for their dipole moments and enforcing the spatial inversion symmetry on the CuCl₄ framework. Both (*R*)- and (*S*)-(MPA)⁺ molecules are simultaneously present in pairs within the unit cell and are related by inversion symmetry (see Supplementary Note 3 and Fig. 13 for more details on the computational methods).** The symbol $\lambda \in [0,1]$ was introduced to **parametrize the chirality-preserving transition path** from the non-centrosymmetric non-polar reference structure (P_0 , $\lambda = 0$) to the polar ground state structure (P_+ or P_- , $\lambda = \pm 1$)⁴⁷, as illustrated in **Fig. 4a.....**”

“.....The magnitude of ξ further quantifies the extent of the framework chirality through its magnitude; therefore, it appears as an indicator of the chirality transfer in this structure. **In the case**

of racemic structure, constructed by imposing inversion symmetry in the unit cell, the system is centrosymmetric and \mathbf{p} vanishes, hence, $\xi = 0$, consistently with the absence of chirality transfer.”

Reviewer: 2

Comments:

In this article, the authors rediscovered the ferroelectricity present in the previously reported compound described in Chem. Mater. 2020, 32, 20, 8914 – 8920, and proposed a new mechanism for chiral transfer. Through Landau symmetry mode analysis, the authors analyzed the cross-coupling of chirality with ferroic orders and proposed a new hybrid improper chirality transfer mechanism, for which the coupling of two non-chiral distortions enables the transfer of chirality from the organic cations to the framework and proposed a new pseudo-scalar quantity, $\xi = \mathbf{p} \cdot \mathbf{r}$ to describe the coupling. Additionally, the authors supplemented the study with MCD (Magnetic circular dichroism) measurements and other relevant tests on the compound.

1. Can ferroelectric and magnetic properties be coupled in the compound? If conditions permit, can the flipping of ferroelectric domains be controlled by a magnetic field Science 367,671-676(2020)?

Response: Thank you for these very constructive comments. By performing Landau symmetry mode analysis (based on an undistorted parent framework with ‘parent’ space group $Cmmm$, #65), this work, **for the first time**, demonstrates that the symmetry-allowed terms in the free energy allow the coherent cross-coupling of ferroelectric and magnetic orders through the intermediation of organic-to-inorganic chirality transfer. We found that the molecular chirality Γ_1^- has an **invariant trilinear coupling** in the free energy to the non-chiral ferroelectric moment Γ_4^- and a non-chiral ferro-rotational bond order Γ_4^+ , providing a novel **hybrid-improper mechanism** for chirality transfer from the organic molecules to the framework. The degree of chirality transfer can be described by the pseudo-scalar quantity $\xi = \mathbf{p} \cdot \mathbf{r}$ (where \mathbf{p} is the ferroelectric displacement vector and \mathbf{r} is the ferro-rotational vector), which differentiates the *R*- and *S*-enantiomers by its sign. This trilinear coupling is accompanied by a **Jahn-Teller antiferrodistortion** (R_1^+) in (*R*)- and (*S*)-chiral perovskites, which is essential to the ordering of **ferromagnetic moments** ($m\Gamma_4^+$,

$m\Gamma_3^+$ and $m\Gamma_2^+$). As a result, the transfer of organic-to-inorganic chirality in this chiral multiferroic perovskites could facilitate a coherent coupling between the in-plane ferroelectric and ferromagnetic orders.

However, experimental evidence of ferroelectric domains modulated by external magnetic fields, such as that of Long *et al.* (*Science* **367**, 671-676 (2020)), is difficult to observe **at room temperature** in this compound because of the relatively low Curie temperature and weak ferromagnetic spin coupling. Magnetoelectric coupling may occur at low temperatures, but requires high-precision low-temperature characterization equipment and a strict experimental design to exclude experimental artifacts. This is indeed what we are exploring now, and it is beyond the scope of this paper. We aim for the current article to present novel insights into the magnetoelectric coupling mechanism from a theoretical perspective and provide the theoretical guidelines for future experimental verification.

In the relevant sections of the manuscript, we now make it clear that symmetry-allowed invariants in the free energy suggest the possibility of novel magnetoelectric phenomena:

Abstract: “..... This mechanism involves the coupling of non-chiral distortions, characterized by a newly defined pseudo-scalar quantity, $\xi = \mathbf{p} \cdot \mathbf{r}$ (\mathbf{p} represents the ferroelectric displacement vector and \mathbf{r} denotes the ferro-rotational vector), which distinguishes between (*R*)- and (*S*)-chiralities based on its sign. Moreover, the reversal of this descriptor's sign can be associated with coordinated transitions in ferroelectric distortions, Jahn-Teller antiferro-distortions, and Dzyaloshinskii-Moriya vectors, indicating a mediating role of crystallographic chirality in magnetoelectric correlations.”

Introduction: “..... This new chirality transfer mechanism can be parametrized by a pseudo-scalar order parameter, $\xi = \mathbf{p} \cdot \mathbf{r}$, (\mathbf{p} is the ferroelectric moment and \mathbf{r} is the ferro-rotational moment, which are non-chiral order parameters), wherein the sign of ξ manifests as +1 and -1 for the (*R*)- and (*S*)-chiral HOIPs, respectively. In addition, the chirality change is allowed by symmetry to couple with J-T pseudo-rotations of the associated orbital ordering as well as the D-M vectors, thereby allowing the possibility of a synergetic correlation between the intralayer ferroelectric and

ferromagnetic moments^{21,22}. This study highlights, for the first time, the correlation of crystallographic chirality with the ferroic behaviors in HOIPs, paving the ground for studying other spin-related properties such as chiral spin textures and chiral magneto-optical effects.”

Above Figure 5: “Additionally, the trilinear coupling of Γ_4^- , Γ_4^+ , and Γ_1^- is accompanied by a J-T-type antiferrodistortion in (*R*)- and (*S*)-chiral HOIPs (**Fig. 5c**), denoted as R_1^+ , which is essential to the intra-layer ferromagnetic orders ($m\Gamma_4^+$, $m\Gamma_3^+$ and $m\Gamma_2^+$). The interplay between J-T distortion and the SOC effect is directly linked to the orbital magnetic moments of HOIPs, where switching the orientation of the J-T distortion interconverts the orthogonality of magnetic $\text{Cu}^{2+} 3d$ orbital ordering. This observation is consistent with the chirality-induced interchange of D-M vectors and the mirrored charge density patterns confirmed by theoretical calculations. This presents a possible scenario where the transfer of chirality from the organic molecules to the inorganic octahedrons in HOIPs facilitates a coherent coupling between the in-plane ferroelectric and ferromagnetic orders (Supplementary Note 4 for more details), which has yet to be experimentally proven.

Conclusion: “In this study, we have presented experimental evidence of chiral multiferroic properties in layered copper-based HOIPs (*R/S*)-(MPA)₂CuCl₄. These materials exhibit in-plane ferroelectric behavior, A-type antiferromagnetic configuration, and chirality-dependent MCD characters. Through the application of Landau symmetry-mode analysis, we have identified a trilinear coupling mechanism that links molecular chirality with the non-chiral ferroelectric and ferro-rotational moments of the inorganic framework. This hybrid-improper mechanism enables the transfer of chirality from the organic molecules to the inorganic framework. The extent of chirality transfer and the differentiation between the (*R*)- and (*S*)- enantiomers can be quantified by the pseudo-scalar quantity $\xi = \mathbf{p} \cdot \mathbf{r}$.

Furthermore, the symmetry-allowed couplings between framework chirality, J-T antiferrodistortion, D-M vectors, bond rotations, and intra-layer ferroelectric and ferromagnetic orders introduce the alluring possibility of observing novel chirality-assisted magnetoelectric phenomena. By leveraging the interplay between chirality and various ferroic orders, we can unlock

new avenues for designing and manipulating materials with enhanced properties and tailored functionalities in the field of quantum technologies.”

To offer further insights into the magnetoelectric coupling mechanism, we have added a new section 4.6 to SI to explore possible magneto-electric domain switching:

“4.6 Possible magneto-electric coupling

Supplementary Table 7 lists the IRREPs and order-parameter directions of topological framework space group $Cmmm$ to which each vector component of the Cu^{2+} electric and magnetic moments. The ferroelectric component $\Gamma_4^-(c)$ along the in-plane parent- Y axis is substantial, whereas the $\Gamma_2^-(d)$ component along the in-plane parent- Z axis and the $\Gamma_3^-(e)$ component along the out-of-plane parent- X axis are either zero or very small. Due to the weak inter-layer magnetic coupling and soft ferromagnetic response, it is appropriate to use ferromagnetic $k = (0,0,0)$ IRREPs to describe the intra-layer ferromagnetic ordering. The A-type antiferromagnetic inter-layer order is actually governed by the corresponding $k = (1,0,0)$ IRREPs which are labeled as Y . This substitution does not affect the free-energy invariants so long as ferro- and anti-ferromagnetic order parameters are not mixed. The in-plane magnetic moment is large, though due to very weak in-plane magnetic anisotropy, that moment can be divided arbitrarily between the $m\Gamma_4^+(f)$ component along the parent- Y axis and the $m\Gamma_2^+(g)$ component along the parent- Z axis. Because the magnetic moment is only weakly canted towards the out-of-plane parent- X axis, the $m\Gamma_3^+(h)$ component is small. Because J-T distortion $R_1^+(0; i)$ is essential for stabilizing the magnetic moments, we can also expect it to play some role in any magneto-electric phenomena.

To explore possible magneto-electric domain switching, all of the unfactorable multi-linear invariants up to order six in these 11 order parameters were identified, each of which is invariant with respect to operations of parent space group $Cmmm$ (i.e. those domain transformations listed in Supplementary Table 8) and cannot be factored into simpler invariant terms. If we select only those multi-linear terms that couple electric and magnetic moments, and ignore terms containing order parameters deemed to be *small*, an invariant of the form $acfgi^2$ emerges as the most promising candidate. This invariant is linear in the chiral order parameter, the ferroelectric moment,

and both in-plane components of the magnetic moment, but quadratic in the amplitude of the J-T distortion. If the $acfgt^2$ term has a substantially non-zero coefficient in the free energy, and if the magnetic moment has a general in-plane direction, this term could in principle facilitate magneto-electric switching phenomena involving exactly one of the two in-plane components of the magnetic moment. Furthermore, the sign of such a magneto-electric coupling will be opposite for the (*R*) and (*S*)-chiralities. If invariants that involve some of the smaller order parameters are considered, an even richer variety of magneto-electric possibilities emerges.”

2. As mentioned by the authors in *Chem. Mater.* 2020, 32, 20, 8914 – 8920, the presence of magneto-chiral dichroism (MChD) in this structure has already been reported, indicating the existence of MCD in the compound. The authors supplemented the MCD spectra and conducted comparisons. How does this g_{mcd} value compare to the devices already in use?

Response: We greatly appreciate the reviewer’s kind suggestion. We have added the field-dependent absorbance and the dissymmetry factor for magneto-chiral dichroism (g_{MChD}) at room temperature in the revised supplementary information. The spectra show the symmetry for (*R/S*)-(MPA)₂CuCl₄. We calculated them using:

$$g_{\text{MChD}} = 2 \times \frac{A(B \uparrow \uparrow \kappa) - A(B \uparrow \downarrow \kappa)}{A(B \uparrow \uparrow \kappa) + A(B \uparrow \downarrow \kappa)}$$

in which, $A(B \uparrow \uparrow \kappa)$ and $A(B \uparrow \downarrow \kappa)$ are the corresponding absorbance for the field in the parallel ($\uparrow \uparrow$) and antiparallel ($\uparrow \downarrow$) with respect to the light propagation direction (κ). We summarized their corresponding g_{MChD} values in the Supplementary Table 3. They are smaller compared to the values given in the literature (Sun *et al. Chem. Mater.* **32**, 8914–8920 (2020)). The difference can be ascribed to the different sample forms, measuring temperatures, and magnetic field strengths. They used the manual grinding powder-form crystals and conducted measurements with an applied magnetic field of **B = 1 T** at **2 K**. By contrast, we used the spin-coating polycrystalline thin films and measured under a magnetic field of **B = 1.6 T** at **300 K**. Nevertheless, ours are still higher than those reported values for the **Eu((±)tfc)₃ complex** ($5 \times 10^{-3} T^{-1}$, Rikken *et al. Nature* **390**, 493-494 (1997)) and the chiral **Ni** nanomagnets ($7.3 \times 10^{-4} T^{-1}$, Eslami *et al. ACS Photonics* **1**, 1231-1236 (2014)).

To further clarify this point, we have added the discussion MChD in the manuscript:

“.....A recent study on MChD demonstrated that (R/S)-(MPA)₂CuCl₄ exhibit pronounced and mirror-imaged MChD signals at 2 K, highlighting a strong correlation between crystallographic chirality and intrinsic magnetism¹⁵. We characterized the dissymmetry factor of MChD (g_{MChD}) for polycrystalline thin films (Supplementary Fig. 4). The corresponding g_{MChD} values are summarized in Supplementary Table 3. The MChD effect stems from the interference of electric dipole and magnetic dipole transitions, as well as the Cotton-Mouton effect due to the second-order perturbation of the magnetic field between excited states²⁶. The values are higher than those reported for Eu((±)tfc)₃ complex ($5 \times 10^{-3} \text{ T}^{-1}$)²⁷ and chiral Ni nanomagnets ($7.3 \times 10^{-4} \text{ T}^{-1}$)²⁸.

REFERENCES

15 Sun, B. *et al.* Two-dimensional perovskite chiral ferromagnets. *Chem. Mater.* **32**, 8914-8920 (2020).

26 Nakagawa, N. *et al.* Magneto-chiral dichroism of CsCuCl₃. *Phys. Rev. B* **96**, 121102 (2017).

27 Rikken, G. L. J. A. & Raupach, E. Observation of magneto-chiral dichroism. *Nature* **390**, 493-494 (1997).

28 Eslami, S. *et al.* Chiral nanomagnets. *ACS Photonics* **1**, 1231-1236 (2014).”

We also added the experimental results of field-dependent absorbance spectra and the dissymmetry factor of magneto-chiral dichroism (g_{MChD}) in **SI** as follows:

Supplementary Fig. 4. (a) Field-dependent absorbance spectra of (R)- and (S)-(MPA)₂CuCl₄ at

room temperature. (b) The dissymmetry factor of magneto-chiral dichroism (g_{MChD}) for (R)- and (S)-(MPA)₂CuCl₄. All the measurements were conducted in ambient conditions.

λ (nm)	230	280	330	390	430
(R)-(MPA) ₂ CuCl ₄ / g_{MChD} ($\times 10^{-3} \text{ T}^{-1}$)	16	7	8	4	6
(S)-(MPA) ₂ CuCl ₄ / g_{MChD} ($\times 10^{-3} \text{ T}^{-1}$)	11	7	14	5	10

Supplementary Table 3. Extracted absolute values of g_{MChD} from the Supplementary Fig. 4 at specific wavelengths.”

- The authors mentioned that 'The effect originates from the fine-tuning of the chiral-dependent exciton energies by the field-induced Zeeman splitting', where the applied magnetic field induces perturbations in the states of transition dipoles (ground, excited, and other energy states), resulting in a universally present and temperature-independent MCD. How did the authors eliminate this interference?

Response: We would like to thank the reviewer’s careful reading of our manuscript and the useful comment. We agree with the reviewer that the interference of the electric and magnetic dipole moment occurs in MChD. In the main content of the manuscript, we do not eliminate this interference. Instead, the magnetic field-induced perturbation toward the electronic states is the origin of the observed MChD (MCD) signal. Here, MCD is associated with the field-induced **Zeeman splitting** at photoexcited states and MChD originates from the **interplay** between **chirality** and **magnetic fields**. The interference between electric dipole (E_1) and magnetic dipole transitions (M_1) is the intrinsic nature of a material. It has been discussed in detail in the previous papers and we also copied here (Sun *et al. Chem. Mater.*, **32**, 8914 (2020); Pan *et al. Chem. Mater.* **35**, 1667-1673 (2023); Han *et al. Adv. Mater.* **32**, 1801491 (2020); Nakagawa *et al. Phys. Rev. B*, **96**, 121102(R) (2017)):

“.....In the UV–vis region, the MChD is generally understood as an interference effect between electric dipole (E_1) and magnetic dipole transitions (M_1) resulting from the SOC. On the basis of these mechanisms, the MChD signal of the $d-d$ transition for Cu^{2+} is expressed by the summation

of the E_1-M_1 interference term of optical transition from the Cu^{2+} ions (Sun *et al. Chem. Mater.*, **32**, 8914 (2020).....”

On the other hand, in this work, we focus on **MCD** rather than MChD spectra, which are two different concepts. In order to better understand their optical origins, we introduce the concept of complete dielectric function (ϵ_{\pm}) (the sign \pm indicates left and right chirality), which can be expressed by an equation as follows (Nakagawa *et al. Phys. Rev. B*, **96**, 121102(R) (2017)):

$$\epsilon_{\pm} = \epsilon_0 \pm \alpha_{\text{NCD}}k_z \pm \beta_{\text{MCD}}M_z + \gamma_{\text{MChD}}k_zM_z$$

Herein, ϵ_0 is the field-independent dielectric constant. α_{NCD} is concerned with the natural optical activity or natural circular dichroism (NCD) originating from the space-inversion symmetry breaking in chiral materials. It also refers to the nonreciprocal directional dichroism (NDD). β_{MCD} is related to MCD due to time-reversal symmetry breaking. The interference between E_1 and M_1 transitions in MCD spectra arises due to the application of a magnetic field, which induces **Zeeman splitting** of the electronic energy levels. When the space inversion and time reversal symmetry breaking occur simultaneously in the chiral materials, the magneto-chiral dichroism (MChD) is developed (the γ_{MChD} term). MChD which is known to be one type of NDD comes from an interference of electric dipole and magnetic dipole transitions.

As we know, the light-matter interaction involves an optical generation for an oscillating electric (magnetic) dipole moment. In achiral materials, optical absorption (or transmission) does not change by the reversal of light propagation direction. Nevertheless, in multiferroic materials, it is possible to have a decent coupling of electric and magnetic dipole moments through spin-orbit coupling. As a result, optical constants that are associated with absorption and transmission may change with the reversal of light propagation direction (Toyoda *et al. Phys. Rev. Lett.* **115**, 267207 (2015)).

To clarify this point, we have added some discussion with the corresponding reference in the revised manuscript:

“Magnetic circular dichroism. The coexistence of optical activity (NOA) and ferromagnetism of (R/S)-(MPA)₂CuCl₄ allow us to explore magneto-chiroptical effects in these crystals. The dielectric

functions ε_{\pm} (the sign \pm indicates left and right chirality) can be decomposed into several components: the field-independent dielectric function, which does not contribute to the chiroptical effects, the natural circular dichroism (NCD) resulting from the space inversion asymmetry, the magnetic circular dichroism (MCD) arising from the breaking of time-reversal symmetry, and the magneto-chiral dichroism (MChD)^{15,26}. A recent study on MChD demonstrated that (*R/S*)-(MPA)₂CuCl₄ exhibits pronounced and mirror-imaged MChD signals at 2 K, highlighting a strong correlation between crystallographic chirality and intrinsic magnetism¹⁵. We characterized the dissymmetry factor of MChD (g_{MChD}) for polycrystalline thin films (Supplementary Fig. 4). The corresponding g_{MChD} values are summarized in Supplementary Table 3.....

REFERENCE

15 Sun, B. *et al.* Two-dimensional perovskite chiral ferromagnets. *Chem. Mater.* **32**, 8914-8920 (2020).

26 Nakagawa, N. *et al.* Magneto-chiral dichroism of CsCuCl₃. *Phys. Rev. B* **96**, 121102 (2017).”

4. The author mentions that between the layers of the compound, there is a ferromagnetic arrangement and a DM interaction in the helical structure. However, from the structure, it can be observed that the Cu-Cl-Cu exhibits an approximately 180° antiferromagnetic arrangement, which contradicts the Goodenough-Kanamori rules.

Response: We thank the reviewer for these constructive comments. The Goodenough-Kanamori rules provide guidelines for predicting magnetic ordering in transition metal compounds based on the interactions between magnetic ions in a crystal lattice. It states that superexchange interactions are **antiferromagnetic** where the virtual electron transfer is between overlapping orbitals that are **each half-filled**, but they are **ferromagnetic** where the virtual electron transfer is from a **half-filled** to an **empty** orbital or from a **filled** to a **half-filled orbital**. In this compound (*R/S*)-(MPA)₂CuCl₄, the **Jahn-Teller** octahedral elongations of Cu²⁺ (*d_g*) are all in plane in a checkerboard pattern of alternating orthogonal elongation directions (**Figure R2 (a)**). Thus, for an adjacent pair of Cu atoms, a **filled** *d*(*x*²-*y*²) orbital on one Cu atom overlaps a **half-filled** *d*(*z*²) orbital on the other (Sun *et al.* *Chem. Mater.* **32**, 8914 (2020)). The Cu-Cl-Cu bonds are around 167° which are **away from 180°**

in symmetric octahedral arrangements. It is in agreement with the Goodenough-Kanamori rules which dictate that this Jahn-Teller antiferro-distortive arrangements promote **ferromagnetic coupling** rather than antiferromagnetic coupling.

In the context of transition metal-based hybrid perovskites, the effect of Goodenough-Kanamori rule has been systematically studied and it shows a strong correlation with the configuration of inorganic MCl_6 octahedral framework. As shown in **Figure R2**, the hybrid perovskites featuring a **Jahn-Teller distorted** octahedral framework will exhibit intra-layer **ferromagnetic coupling** (Cr^{2+} and Cu^{2+}), while in the case of a **linear symmetric** octahedral framework (Fe^{2+} and Mn^{2+}), the coupling will be **antiferromagnetic** (Xue *et al. Chem. Mater.* **34**, 2813–2823 (2022). Zheng *et al. Adv. Mater.* **36**, 2308051 (2024)). Thus, our (R/S) - $(MPA)_2CuCl_4$ should be ferromagnetically coupled within the $CuCl_6$ plane due to the strong Jahn-Teller distorted inorganic framework. It accords with the Goodenough-Kanamori rule, and its spin arrangements are analogous to the Cu^{2+} - and Cr^{2+} -based metal–organic frameworks (**MOFs**), which have been systematically discussed in the previous reports (Stroppa *et al. Adv. Mater.* **25**, 2284-2290 (2013); Stroppa *et al. Angew. Chem. Int. Ed.* **50**, 5847-5850 (2011)). The corresponding discussion has been copied as follows:

*“.....The Goodenough–Kanamori–Anderson (GKA) rules suggest that there is a strong antiferromagnetic coupling if singly occupied orbitals on corresponding sites are directed towards each other. If, however, an occupied orbital is directed towards an empty (or doubly occupied) one, there will be a weaker ferromagnetic coupling. Thus, in our case, the antiferro-orbital ordered planes are coupled ferromagnetically, while they order antiferromagnetically (AFM) in the perpendicular direction, that is, the ground state displays an A-type AFM spin configuration (Stroppa *et al. Angew. Chem. Int. Ed.* **50**, 5847-5850 (2011)).”*

[Redacted]

Figure R2. Schematic diagrams that demonstrate the effect of Goodenough-Kanamori rule in transition metal-based hybrid perovskites. (Xue *et al. Chem. Mater.* **34**, 2813–2823 (2022); Zheng *et al. Adv. Mater.* **36**, 2308051 (2024))

5. In line 218, the authors mentioned the ferroelectric-to-paramagnetic phase transition.

Considering the context, should it be interpreted as ferroelectric-to-paraelectric phase transition?

Response: Thank you so much for your careful review and sorry for the typo. Yes, it should be **ferroelectric-to-paraelectric transition** in the context and this sentence has been revised as follows:

“The absence of ferroelectric-to-**paraelectric** phase transition up to melting points T_M of 450 K is observed in the thermogravimetric analysis (TGA) and differential scanning calorimetry (DSC) measurements.....”

6. In Table 1, $P1$, as a non-centrosymmetric polar space group, should provide the Flack parameter.

Response: We thank the reviewer for clarifying this point. The Flack parameters were added in the last row of the single-crystal XRD forms. We also supplemented the single-crystal XRD data as

well as the corresponding Flack parameters at 150 K, 200 K, and 300 K in the **Supplementary Tables 1 and 2** as follows:

“

(R)-(MPA)₂CuCl₄				
Empirical formula	C₁₈H₂₈Cl₄CuN₂	C₁₈H₂₈Cl₄CuN₂	C₁₈H₂₈Cl₄CuN₂	C₁₈H₂₈Cl₄CuN₂
Formula weight	477.76 g/mol	477.76 g/mol	477.76 g/mol	477.76 g/mol
Temperature/K	100.00 K	150.00 K	200.00 K	300.00 K
Crystal system	triclinic	triclinic	triclinic	triclinic
Space group	P 1	P 1	P 1	P 1
a/Å	7.5711(3) Å	7.5916(4) Å	7.6137(3) Å	7.6655(3) Å
b/Å	7.5732(3) Å	7.5949(3) Å	7.6180(2) Å	7.6684(3) Å
c/Å	17.8700(7) Å	17.8982(9) Å	17.9298(6) Å	17.9969(7) Å
α/°	80.8830(10)°	80.926(2)°	80.9850(10)°	81.1260(10)°
β/°	86.802(2)°	86.753(2)°	86.6570(10)°	86.4910(10)°
γ/°	89.4090(10)°	89.3770(10)°	89.3960(10)°	89.4380(10)°
Volume/Å ³	1010.10(7) Å³	1017.41(8) Å³	1025.35(6) Å³	1043.27(7) Å³
Z	2	2	2	2
Density (calculated)	1.571 g/cm³	1.560 g/cm³	1.547 g/cm³	1.521 g/cm³
Absorption coefficient	1.615 mm⁻¹	1.603 mm⁻¹	1.591 mm⁻¹	1.563 mm⁻¹
F(000)	494.0	494.0	494.0	494.0
2θ range for data collection/°	4.624 to 61.066	4.616 to 61.024	4.608 to 61.122	4.59 to 61.06
Reflections collected	56229	54074	55860	54316
Data/restraints/parameters	12271/95/528	12336/95/528	12490/95/528	12700/95/528
Goodness-of-fit on F ²	1.043	1.021	1.033	1.030
Final R indexes [I>2σ (I)]	R₁ = 0.0270, wR₂ = 0.0594	R₁ = 0.0258, wR₂ = 0.0594	R₁ = 0.0294, wR₂ = 0.0628	R₁ = 0.0355, wR₂ = 0.0747
Flack parameter	0.001(4)	-0.003(3)	-0.003(4)	-0.002(5)

Supplementary Table 1. Single-crystal XRD results for **(R)-(MPA)₂CuCl₄**.

(S)-(MPA)₂CuCl₄				
Empirical formula	C₁₈H₂₈Cl₄CuN₂	C₁₈H₂₈Cl₄CuN₂	C₁₈H₂₈Cl₄CuN₂	C₁₈H₂₈Cl₄CuN₂
Formula weight	477.76 g/mol	477.76 g/mol	477.76 g/mol	477.76 g/mol
Temperature/K	100.00 K	150.00 K	200.00 K	300.00 K
Crystal system	triclinic	triclinic	triclinic	triclinic
Space group	P 1	P 1	P 1	P 1
a/Å	7.5876(3) Å	7.5917(3) Å	7.6164(2) Å	7.6652(3) Å
b/Å	7.5928(3) Å	7.5934(3) Å	7.6171(3) Å	7.6741(2) Å
c/Å	17.9130(7) Å	17.8945(7) Å	17.9279(6) Å	18.0025(7) Å
α/°	80.8960(10)°	80.9730(10)°	81.0150(10)°	81.1800(10)°
β/°	86.8300(10)°	86.7510(10)°	86.6890(10)°	86.532(2)°
γ/°	89.3750(10)°	89.3910(10)°	89.3970(10)°	89.4260(10)°
Volume/Å ³	1017.43(7) Å³	1017.15(7) Å³	1025.61(6) Å³	1044.53(6) Å³
Z	1	2	2	2
Density (calculated)	1.560 g/cm³	1.560 g/cm³	1.547 g/cm³	1.519 g/cm³
Absorption coefficient	1.603 mm⁻¹	1.604 mm⁻¹	1.590 mm⁻¹	1.562 mm⁻¹
F(000)	494.0	494.0	494.0	494.0
2θ range for data collection/°	4.612 to 60.974°	4.616 to 61.012°	4.608 to 61.008°	4.588 to 61.004°
Reflections collected	66898	74460	77766	59409
Data/restraints/parameters	12244/95/504	12367/95/504	12466/94/504	12714/94/504
Goodness-of-fit on F ²	1.042	1.024	1.046	1.037
Final R indexes [I>=2σ (I)]	R₁ = 0.0274, wR₂ = 0.0595	R₁ = 0.0308, wR₂ = 0.0644	R₁ = 0.0298, wR₂ = 0.0614	R₁ = 0.0384, wR₂ = 0.0772
Flack parameter	-0.010(3)	-0.004(4)	-0.003(4)	-0.003(6)

Supplementary Table 2. Single-crystal XRD results for **(S)-(MPA)₂CuCl₄.**”

The authors sincerely appreciate the constructive comments from the reviewers, which have been crucial in improving the quality of our work. Each point raised has been carefully addressed, and necessary revisions have been made to improve the clarity and scientific rigor of the paper. We hope that our detailed responses and improvements meet the expectations, and that the manuscript will be considered for publication.

REVIEWERS' COMMENTS

Reviewer #1 (Remarks to the Author):

I have carefully examined the response letter and the revisions made by the authors. I feel that the authors have effectively addressed all my concerns, including the issues of novelty and phase transition. Despite the difficulty in obtaining the racemic crystal, the authors have clearly stated this challenge. Additionally, they have provided a clear response and revision to reviewer 2's questions. The content of the paper has been significantly improved compared to the previous version. Therefore, I suggest the publication of this paper with no further revision.

Reviewer #2 (Remarks to the Author):

The authors have revised the paper as I suggested, so please publish as is.

Point-to-Point Response to Reviewers' Comments

The reviewers' comments are marked in **black**, and the author's responses are in **blue**.

Reviewer: 1

Comments:

I have carefully examined the response letter and the revisions made by the authors. I feel that the authors have effectively addressed all my concerns, including the issues of novelty and phase transition. Despite the difficulty in obtaining the racemic crystal, the authors have clearly stated this challenge. Additionally, they have provided a clear response and revision to reviewer 2's questions. The content of the paper has been significantly improved compared to the previous version. Therefore, I suggest the publication of this paper with no further revision.

Response: We would like to thank the reviewer for the high evaluation and valuable suggestions.

Reviewer: 2

Comments:

The authors have revised the paper as I suggested, so please publish as is.

Response: Thank you so much for your suggestions and comments.